# Theoretical and Numerical Study of Self-Organizing Processes in a Closed System Classical Oscillator and Random Environment

Ashot S. Gevorkyan [1,2,*], Aleksander V. Bogdanov [3,4], Vladimir V. Mareev [3] and Koryun A. Movsesyan [1]

1    Institute for Informatics and Automation Problems NAS of RA, 1, P. Sevak Str., Yerevan 0014, Armenia
2    Institute of Chemical Physics, NAS of RA, 5/2, P. Sevak Str., Yerevan 0014, Armenia
3    Faculty of Applied Mathematics and Control Processes, St. Petersburg State University , 7/9 Universitetskaya nab., 199034 St. Petersburg, Russia
4    Center for Advanced Digital Technologies, St. Petersburg State Marine Technical University, Lotsmanskaya D. 3, 190121 St. Petersburg, Russia
*    Correspondence: g_ashot@sci.am

**Abstract:** A self-organizing joint system classical oscillator–random environment is considered within the framework of a complex probabilistic process that satisfies a Langevin-type stochastic differential equation. Various types of randomness generated by the environment are considered. In the limit of statistical equilibrium (SEq), second-order partial differential equations (PDE) are derived that describe the distribution of classical environmental fields. The mathematical expectation of the oscillator trajectory is constructed in the form of a functional-integral representation, which, in the SEq limit, is compactified into a two-dimensional integral representation with an integrand: the solution of the second-order complex PDE. It is proved that the complex PDE in the general case is reduced to two independent PDEs of the second order with spatially deviating arguments. The geometric and topological features of the two-dimensional subspace on which these equations arise are studied in detail. An algorithm for parallel modeling of the problem has been developed.

**Keywords:** general theory of random and stochastic dynamical systems; partial differential equations; measure and integration; noncommutative differential geometry; parallel computing

## 1. Introduction

From ancient sources, we know that Pythagoras (l. c. 571–497 BC) and the students of his school considered complex numerical and geometric constructions, as well as complicated natural phenomena. Nonetheless, the metaphorization of the word "system" seems to have been first proposed by Democritus (born c. 460 BCE—died c. 370), which meant the formation of complex bodies from atoms, similar to the formation of words from syllables and syllables from letters. In addition, in ancient Greek philosophy, the "system" characterized the orderliness and integrity of natural objects. Sometime later, Plato (427–347 BC) formulated the thesis that *the whole is greater than the sum of its parts*. Aristotle (384–322 BC), being in a polemic with Plato, formulated the opposite thesis, saying that the whole can be decomposed and studied separately, and then put back together again without losing anything (see for example [1,2]). In those distant times, the Aristotelian concept became more popular due to its simplicity, and for the next almost 2500 years, all research on the problems of natural science was carried out within the framework of this particular concept. Despite its idealized nature, this concept remained the main method of cognition in science for a long time, stimulated its development and contributed to the creation of a huge number of new technologies. Nevertheless, the triumph of Aristotle's concept at the beginning of the 20th century was the creation of a logically perfect theory—the classical

mechanics of closed systems, which after a short time experienced a deep crisis, which led to the development of a more general physical representation of quantum mechanics.

In the 20th century, Einstein and Smoluchowski developed the theory of Brownian particle motion [3,4], which stimulated the emergence of a new approach in the study of nature—the theory of classical and quantum open systems [5]. Recall that an open physical system, interacting with the environment, unlike an isolated system, exchanges mass, energy, information, etc. with it. At present, the science of open systems is intensively developing both theoretically and experimentally, especially in the fields of modern quantum physics, chemical physics, etc. [6–8]. Note that the open systems approach is not equivalent to Plato's concept, but it is closer to it in spirit. Nevertheless, as shown by numerous studies, this approach also has serious difficulties that do not allow one to describe a number of important phenomena in nonequilibrium thermodynamics, many-particle quantum systems, etc. Recall that the main drawback of all representations of open systems is that when describing various physical processes, a certain part of information is inevitably lost, especially when it comes to physical systems under extreme conditions. This is due to ignoring the influence of the system on the medium, which excludes the possibility of formation of a *small environment* (SE) self-consistent with it, which, most likely, can be considered as its integral part or continuation [9].

The purpose of this study is to combine two opposite concepts of nature cognition, namely, to supplement an open system with its environment and to describe the *joint system* JS as a closed system. Note that just such a statement of the problem would be equivalent to Plato's concept!

In this paper, we demonstrate the implementation of this idea on the example of the well-known problem of a classical oscillator immersed in a random environment and under the action of an external force. Recall that this problem has been studied in sufficient detail within the framework of various models of Brownian motion [10], and its results are widely used in solving a number of important applied problems (see [11]). In particular, a Brownian particle moving in a viscous medium exchanges energy and momentum with the environment, which affects the motion of the particle itself. In other words, such mutual influence leads to the appearance of memory in the Brownian particle, i.e., its behavior becomes dependent on the entire previous history of the process. However, we know that a Markov model, by definition, cannot describe random processes with memory, and therefore, taking into account the entrainment of particles of the medium imparts a non-Markov character to the Brownian motion. To overcome the difficulties of describing non-Markovian Brownian motion, it is necessary to reformulate the standard statement of the problem, making significant changes to the mathematical apparatus. In connection with this, a number of authors have proposed the so-called generalized Langevin equation, in which instead of a resistance force proportional to velocity, an integral operator of the convolution type is used [12]. Note that despite the adaptation of the theory of Brownian motion to emerging new scientific and technical problems, this is the theory of open systems and has all the above limitations and disadvantages. In other words, the standard theories of Brownian motion are unsuitable for describing the properties of systems far from equilibrium or in critical states.

Thus, the solution to the problem lies in the development of a fundamentally new mathematical representation, which makes it possible to study the process of self-organization of the entire system, consisting of finite and infinite subsystems, as a closed system. The article is organized as follows:

In Section 2, we present a statement of the problem and derive complex stochastic differential equations describing the movement of fields of a random environment for three different cases:

- The oscillator frequency is random and there is no external field;
- The oscillator frequency is random and the external field is a regular function;
- The frequency of the oscillator is a regular function and the external force is random.



Note that for all three cases, we have obtained the distribution equations for the fields of the environment using the systems of the corresponding stochastic differential equations.

In Section 3, we represent in detail a method for constructing a function space measure. Formulas are derived for the mathematical expectation of the oscillator trajectory in the form of a double integral representation for the three cases indicated above, where the integrand is a solution of the corresponding complex second-order *partial differential equation* PDE.

In Section 4, we obtain a 4th order algebraic equation with coefficients depending on two variables and time, which generates a two-dimensional compactified subspace on which second-order complex PDEs are defined. The geometric and topological features of a two-dimensional manifold are studied in detail. We prove that the formation of topological manifolds with the first Betti [13] number is less than or equal to 4, depending on the interaction constants of the oscillator with the environment.

In Section 5, we present a complex second-order PDE as a system of two coupled real PDEs and formulate the Neumann initial-boundary value problem for this system. We analyze in detail the PDEs system for the case of symmetry or asymmetry as well as the absence of any symmetry of the desired solutions. Using the symmetry properties of the problem, we proved that in the case of symmetry or antisymmetry of solutions, the system of equations reduces to two independent PDEs. Finally, we have shown that when the solutions do not have a certain symmetry, the PDEs system again reduces to two independent PDEs but with deviant arguments.

In Section 6, we construct the time-dependent Shannon entropy for a classical oscillator as an open system affected by the environment. In the same section, we constructed the generalized Shannon entropy for a closed self-organizing system oscillator and random environment. In Section 7, we develop algorithms for numerical simulation of the PDE system as well as analyze, interpret, and visualize the results of various numerical experiments. In Section 8, we discuss the obtained theoretical and numerical results and outline directions for future research.

## 2. Problem

### 2.1. Statement of the Problem

The classical action of a one-dimensional oscillator immersed in a random environment can be represented as (see [14]):

$$\mathbb{S}[x, t_i, t_f] = \int_{t_i}^{t_f} \mathbb{L}(\dot{x}(t), x(t), t) dt, \tag{1}$$

where $t_i$ and $t_f$ are the moments of time when the interaction of the oscillator with the environment turns on and off, respectively; in addition, $\mathbb{L}(\dot{x}(t), x(t), t)$ is the Lagrangian describing the oscillator with a random environment:

$$\mathbb{L}(\dot{x}(t), x(t), t) = \frac{1}{2}\dot{x}^2 - \frac{1}{2}\Omega^2(t; \{\mathbf{f}\}) x^2 + F(t; \{\mathbf{g}\}) x. \tag{2}$$

Recall that $\{\mathbf{f}\}$ and $\{\mathbf{g}\}$ are complex probabilistic processes (stochastic sources or generators), whose properties will be refined below. Obviously, the presence of stochastic generators in the Lagrangian makes the action also stochastic. Despite this, one can still require the fulfillment of the minimization condition:

$$\delta\mathbb{S}[x, t_i, t_f] = \delta \int_{t_i}^{t_f} \mathbb{L}(\dot{x}(t), x(t), t) dt = 0. \tag{3}$$

Performing the standard procedure for varying the expression (3), taking into account the conditions $\delta x(t_i) = 0$ and $\delta x(t_f) = 0$ (see [15]), we obtain the following second-order

differential equation describing the motion of a test particle, i.e., classical oscillator in a random environment (thermostat):

$$\ddot{x} + \Omega^2(t; \{\mathbf{f}\})x = F(t; \{\mathbf{g}\}), \qquad x, t \in (-\infty, +\infty), \tag{4}$$

where $\dot{x} = dx/dt$.

It is important to note that the randomness of the action $\mathbb{S}[x, t_i, t_f)$ and, accordingly, the Lagrangian $\mathbb{L}(\dot{x}(t), x(t), t)$ does not affect the variation procedure, as a result of which a stochastic Equation (4) is found. However, as we well know, the second-order Equation (4) is still undefined, since the stochastic equation must be first order.

For definiteness, we will assume that random generators satisfy the *white noise* correlation relations:

$$\mathbb{E}\big[f^{(v)}(t)\big] = 0, \qquad \mathbb{E}\big[f^{(v)}(t)f^{(v)}(t')\big] = 2\epsilon_f^{(v)}\,\delta(t - t'),$$

$$\mathbb{E}\big[g^{(v)}(t)\big] = 0, \qquad \mathbb{E}\big[g^{(v)}(t)g^{(v)}(t')\big] = 2\epsilon_g^{(v)}\,\delta(t - t'), \quad v = (i, r), \tag{5}$$

where $\mathbb{E}[...]$ denotes the expectation of the random variable, the set of random generators $\{f^{(r)}, g^{(r)}\}$ and $\{f^{(i)}, g^{(i)}\}$; then, they characterize, respectively, elastic and inelastic collisions of the oscillator with a random environment, while $\{\epsilon_f^{(r)}, \epsilon_g^{(r)}\}$ and $\{\epsilon_f^{(i)}, \epsilon_g^{(i)}\}$ are sets of constants that describe the powers of these processes.

We consider two different cases:

1. When randomness in a JS generates a complex process $\{\mathbf{f}\} \neq 0$, and the second source of the random process is absent $\{\mathbf{g}\} \equiv 0$, and, accordingly,
2. when $\{\mathbf{f}\} \equiv 0$ and randomness in a JS generates the generator $\{\mathbf{g}\} \neq 0$, which has a complex character.

In the case when the external force is an arbitrary regular function of time, i.e., $F_0(t) = F(t; \{\mathbf{g}\})\big|_{\{\mathbf{g}\} \equiv 0}$, the solution of Equation (4) can be formally represented as (see [16]):

$$x(t) = \frac{1}{\sqrt{2\Omega_0^-}}\big[\xi(t)d^*(t) + \xi^*(t)d(t)\big], \qquad d(t) = \frac{i}{\sqrt{2\Omega_0^-}}\int_{-\infty}^t \xi(t')F_0(t')dt', \tag{6}$$

where the symbol "*" denotes the complex conjugation of a function, $\xi(t)$ is the solution of the homogeneous Equation (4), i.e., when $F(t; \{\mathbf{g}\}) \equiv 0$, in addition, the following notations are made; $\Omega_0^- = \lim_{t \to -\infty} \Omega(t, \{\mathbf{f}\}) = const_-$ and $\Omega_0^+ = \lim_{t \to +\infty} \Omega(t, \{\mathbf{f}\}) = const_+$.

Note that in the general case, the asymptotic states $(in)$ at $t \to -\infty$ and $(out)$ at $t \to +\infty$ can be different and, accordingly, $const_- \neq const_+$. Below, for definiteness, we will use the model of the regular frequency $\Omega_0(t)$, which has the following form:

$$\Omega_0(t) = 2 + \frac{1}{\gamma}\big[1 + \tanh(\nu t)\big], \tag{7}$$

where $\gamma, \nu > 0$ are some constants.

### 2.2. Derivation of Environmental Fields Distribution Equations

**Theorem 1.** *If we assume that Equation (4) for the case $F(t; \{\mathbf{g}\}) \equiv 0$ reduces to a complex Langevin SDE, and the complex force $\{\mathbf{f}\}$ is a Gauss–Markovian random process (5), then the conditional probability distribution of the environmental fields in the limit of statistical equilibrium will obey the Fokker–Planck-type equation.*

**Proof.** We can represent the solution of the classical oscillator Equation (4) as:

$$\xi(t) = \begin{cases} \xi_0(t), & t \leq t_0, \\ \xi_0(t_0)\exp\big\{\int_{t_0}^t \phi(t')\,dt'\big\}, & t > t_0, \end{cases} \tag{8}$$

where $\zeta_0(t)$ is the solution of the classical oscillatory Equation (4) in the case when the frequency is a regular function of time $\Omega_0(t) = \Omega(t; \{\mathbf{f}\})|_{\{\mathbf{f}\}\equiv\mathbf{0}}$ and the external regular force is identically equal to zero $F_0(t) \equiv 0$; in addition, $t_0 = 0$ denotes the time of switching on a random environment. As for the function $\phi(t)$, it denotes a complex probabilistic process.

Substituting (8) into (4), taking into account the regular equation:

$$\ddot{\zeta}_0 + \Omega_0^2(t)\zeta_0 = 0, \tag{9}$$

we obtain the following non-linear *stochastic differential equation* (SDE) of Langevin type:

$$\dot{\phi} + \phi^2 + \Omega_0^2(t) + f(t) = 0, \qquad \dot{\phi} = d\phi/dt, \tag{10}$$

where $\Omega^2(t; \{\mathbf{f}\}) = \Omega_0^2(t) + f(t)$.

For further study of the problem, it is convenient to represent the complex probabilistic process $\phi(t)$ as a sum of fields:

$$\phi(t) = u_1(t) + iu_2(t). \tag{11}$$

Using Equation (10) and representation (11), we can write the following system of non-linear SDEs [9]:

$$\begin{cases} \dot{u}_1 = (u_2)^2 - (u_1)^2 - \Omega_0^2(t) - f^{(r)}(t), \\ \dot{u}_2 = -2u_1u_2 - f^{(i)}(t), \end{cases} \tag{12}$$

where $f(t) = f^{(r)}(t) + if^{(i)}(t)$.

Note that the environment fields satisfy the following initial conditions:

$$\dot{u}_1(t_0) = Re\{\dot{\zeta}_0(t_0)/\zeta_0(t_0)\} = 0, \qquad \dot{u}_2(t_0) = Im\{\dot{\zeta}_0(t_0)/\zeta_0(t_0)\} = \Omega_-.$$

Let us consider the following functional describing the distribution of the conditional probability of fields:

$$P(\mathbf{u}, t|\mathbf{u}', t') = \langle \delta[\mathbf{u}(t) - \mathbf{u}(t')]\rangle, \qquad \mathbf{u} = (u_1, u_2). \tag{13}$$

Differentiating expression (13) with respect to the time "$t$", taking into account Equation (10), we obtain:

$$\begin{aligned}
\partial_t P(\mathbf{u}, t|\mathbf{u}', t') &= -\partial_\mathbf{u}\langle \mathbf{u}_t \delta[\mathbf{u}(t) - \mathbf{u}(t')]\rangle = \\
&\partial_\mathbf{u}\{\mathbf{K}(\mathbf{u}, t)P(\mathbf{u}, t|\mathbf{u}', t') + \langle\{f\}\delta[\mathbf{u}(t) - \mathbf{u}(t')]\rangle\},
\end{aligned} \tag{14}$$

where $\mathbf{u}_t = \partial_t\mathbf{u}$ and $\mathbf{u}' \equiv \mathbf{u}(t')$, in addition:

$$\mathbf{K}(\mathbf{u}, t) = \begin{cases} k_1(u_1, u_2, t) = (u_1)^2 - (u_2)^2 + \Omega_0^2(t), \\ k_2(u_1, u_2, t) = 2u_1u_2. \end{cases} \tag{15}$$

Taking into account that the vector probabilistic process $\{\mathbf{f}\}$ satisfies the correlation relations (5), we can calculate the second term in expression (14). Using Wick's theorem for an arbitrary functional $N(\mathbf{u}, t; \{f\})|\mathbf{u}', t')$ of the argument $\{f\}$ (see [17]), we can obtain:

$$\begin{aligned}
\langle\{f\}N(\mathbf{u}, t; \{f\})|\mathbf{u}', t')\rangle &= 2\left\langle\frac{\delta N(t; \{f\})}{\delta f^{(i)}(t)}\right\rangle + 2\left\langle\frac{\delta N(t; \{f\})}{\delta f^{(r)}(t)}\right\rangle \\
&= 2\partial_{u_1}\left\langle\frac{\delta u_1(t)}{\delta f^{(r)}(t)}\delta[\mathbf{u}(t) - \mathbf{u}(t')]\right\rangle + 2\partial_{u_2}\left\langle\frac{\delta u_2(t)}{\delta f^{(i)}(t)}\delta[\mathbf{u}(t) - \mathbf{u}(t')]\right\rangle.
\end{aligned} \tag{16}$$

Since $u_1(t)$ and $u_2(t)$ are stochastic functions, the variational derivatives with respect to the independent random forces $f^{(r)}(t)$ and $f^{(i)}(t)$ can be defined as follows:

$$\left\langle \frac{\delta u_1(t)}{\delta f^{(r)}(t)} \right\rangle = \epsilon_f^{(r)} sgn(t - t') + O(t - t'), \qquad \left\langle \frac{\delta u_2(t)}{\delta f^{(i)}(t)} \right\rangle = \epsilon_f^{(i)} sgn(t - t') + O(t - t'). \quad (17)$$

After carrying out the regularization procedure in the sense of the Fourier expansion, we find its value at the time $t = t'$: $\epsilon_f^{(v)} sgn(0) = \frac{1}{2}\epsilon_f^{(v)}$. Considering the equalities in (17) for the conditional probability, we obtain the following Fokker–Planck equation:

$$\partial_t P = \hat{L}(\mathbf{u}, t)P, \qquad \partial_t \equiv \partial/\partial t, \quad \mathbf{u} = \mathbf{u}(u_1, u_2). \quad (18)$$

In Equation (18), the evolution operator $\hat{L}(\mathbf{u}, t)$ has the form:

$$\hat{L} = \epsilon_f^{(r)} \frac{\partial^2}{\partial u_1^2} + \epsilon_f^{(i)} \frac{\partial^2}{\partial u_2^2} + k_1(u_1, u_2, t)\frac{\partial}{\partial u_1} + k_2(u_1, u_2, t)\frac{\partial}{\partial u_2} + k_0(u_1, u_2, t), \quad (19)$$

where $k_0 = 4u_1$. In addition, in Equations (18) and (19), the variables $u_1$ and $u_2$ denote the coordinates of the distribution of the environmental fields in the state of quasi-equilibrium.

In the case when $t' = t_0$, the conditional probability $P(\mathbf{u}, t) \equiv P(\mathbf{u}, t|\mathbf{u}_0, t_0)$ describes the distribution of classical fields without taking into account the influence of the oscillator on the environment. An important condition for the exact formulation of the problem is the definition of the type of two-dimensional space, on which, let me remind you, the equation for the distribution of environmental fields is set. The latter, in particular, implies writing the conditional probability Equations (18) and (19) in tensor form. □

For simplicity, below, we assume that the Fokker–Planck Equations (18) and (19) are defined on a two-dimensional Euclidean space and solve this equation as Neumann's initial-boundary value problem [9]. The numerical study of the free fields of the environment $P(u_1, u_2, t)$ is carried out using the mathematical algorithm-difference Equation (87) developed in **Listing 1** (Section 7). To illustrate the calculations, graphs of the distribution of fields for various media depending on time are plotted (see Figures 1–3 of Section 7.1).

Now, consider the case when the frequency of the oscillator is regular $\Omega_0(t)$, while the external force, on the contrary, is random and can be represented as the sum:

$$F(t; \{\mathbf{g}\}) = F^{(r)}(t; \{\mathbf{g}\}) + iF^{(i)}(t; \{\mathbf{g}\}) \neq 0, \quad (20)$$

where $F^{(r)}(t; g^{(v)}(t)) = F_0(t) + \sqrt{\epsilon_g^{(r)}}\bar{g}(t)$ and $F^{(i)}(t; g^{(v)}(t)) = \sqrt{\epsilon_g^{(i)}}\bar{g}(t)$; in addition, $\bar{g}(t)$ is a real Gauss–Markov random process, which will be clearly defined below. In particular, using the definition (20) and given that the frequency is regular, we can write the Equation (4) as follows:

$$\ddot{x} + \Omega_0^2(t)x = F_0(t) + \left[\sqrt{\varepsilon^{(r)}} + i\sqrt{\varepsilon^{(i)}}\right]\bar{g}(t). \quad (21)$$

**Theorem 2.** *If the oscillator trajectory obeys Equation (21), and the random function $\bar{g}(t)$ satisfies the Gauss–Markov random process (5), then the distribution of fields of the environment in the limit of statistical equilibrium will be described by PDE of the second order, which, in the (out) asymptotic state or in the limit $t \to +\infty$, transforms into the PDE of the Fokker–Planck type.*

**Proof.** Let us represent the solution of the Equation (21) in the form:

$$x_1(t) = \begin{cases} x_0(t), & t \leq t_0, \\ x_0(t_0)\exp\left\{\int_{t_0}^t \theta(t')\,dt'\right\}, & t > t_0, \end{cases} \quad (22)$$

where $x_0(t)$ is the solution of a classical oscillator with a regular non-stationary frequency under the action of an external non-stationary force and satisfies the following second-order differential equation:

$$\ddot{x}_0 + \Omega_0^2(t)x_0 = F_0(t). \tag{23}$$

Substituting (22) into Equation (21) and assuming that $\theta(t) = w_1(t) + iw_2(t)$, we obtain the following system of stochastic integro-differential equations:

$$\begin{cases} \dot{w}_1 = (w_2)^2 - (w_1)^2 - \Omega_0^2(t) + F_0(t)e^{-\sigma_1(t)}\cos\sigma_2(t) + \bar{g}(t)\mathcal{A}^+(t), \\ \dot{w}_2 = -2w_1w_2 - F_0(t)e^{-\sigma_1(t)}\sin\sigma_2(t) + \bar{g}(t)\mathcal{A}^-(t), \end{cases} \tag{24}$$

where

$$\mathcal{A}^+(t) = \left[\sqrt{\epsilon_g^{(r)}}\cos\sigma_2(t) + \sqrt{\epsilon_g^{(i)}}\sin\sigma_2(t)\right]e^{-\sigma_1(t)},$$

$$\mathcal{A}^-(t) = \left[\sqrt{\epsilon_g^{(i)}}\cos\sigma_2(t) - \sqrt{\epsilon_g^{(r)}}\sin\sigma_2(t)\right]e^{-\sigma_1(t)}.$$

As for the functions $\sigma_1(t)$ and $\sigma_2(t)$, they are singly differentiable, i.e., belong to class $L_1$ and are represented as:

$$\sigma_1(t) = \int_{t_0}^{t} w_1(t')dt' + Re[\ln x_0(t_0)], \qquad \sigma_2(t) = \int_{t_0}^{t} w_2(t')dt' + Im[\ln x_0(t_0)].$$

Assuming that the random function $\bar{g}(t)$ satisfies the white noise correlation relations:

$$\langle \bar{g}(t) \rangle = 0, \qquad \langle \bar{g}(t)\bar{g}(t') \rangle = 2\delta(t - t'),$$

we can use a system of stochastic differential Equations (66) and obtain the following equation for the conditional probability of the environmental fields:

$$\partial_t \mathcal{P} = \hat{\mathcal{L}}(\mathbf{w}, t)\mathcal{P}, \qquad \mathbf{w} = \mathbf{w}(w_1, w_2), \tag{25}$$

where the evolution operator $\hat{\mathcal{L}}$ has the form:

$$\hat{\mathcal{L}}(\mathbf{w}, t) = \mathcal{A}^-(t)\frac{\partial^2}{\partial w_1^2} + \mathcal{A}^+(t)\frac{\partial^2}{\partial w_2^2} +$$

$$h_1(w_1, w_2, t)\frac{\partial}{\partial w_1} + h_2(w_1, w_2, t)\frac{\partial}{\partial w_2} + h_0(w_1, w_2, t), \tag{26}$$

in addition:

$$h_1(w_1, w_2, t) = (w_1)^2 - (w_2)^2 + \Omega_0^2(t) - F_0(t)e^{-\sigma_1(t)}\cos\sigma_2(t),$$

$$h_2(w_1, w_2, t) = 2w_1w_2 + F_0(t)e^{-\sigma_1(t)}\sin\sigma_2(t), \quad h_0(w_1, w_2, t) = 4w_1.$$

Obviously, if the medium is turned on in the time interval $[t_i, t_f]$, then after $t \geq t_f$, Equation (25) becomes the Fokker–Planck-type equation. $\square$

## 3. The Mathematical Expectation of the Trajectory

### 3.1. The Measure of the Functional Space

For further analytical constructions of the theory, it is necessary to determine the distance between functions in the functional space $\mathbb{R}_{\{\xi\}}$ or, more precisely, the measure of the function space (see [18,19]). Let the conditional probability $P(\mathbf{u}, t|\mathbf{u}', t')$ satisfy the following limiting condition:

$$\lim_{t \to t'} P(\mathbf{u}, t|\mathbf{u}', t') = \delta(\mathbf{u} - \mathbf{u}'), \qquad t = t' + \Delta t. \tag{27}$$

The latter means that for small intervals, i.e., for $\Delta t = t - t' \ll 1$, we can represent the solution of Equations (18) and (19) as:

$$P(\mathbf{u}, t | \mathbf{u}', t') = \frac{1}{2\pi \sqrt{|\det \boldsymbol{\epsilon}| \Delta t}} \times$$

$$\exp\left\{ -\frac{[\mathbf{u} - \mathbf{u}' - \mathbf{K}(\mathbf{u}, t)\Delta t]^T \boldsymbol{\epsilon}^{-1} [\mathbf{u} - \mathbf{u}' - \mathbf{K}(\mathbf{u}, t)\Delta t]}{2\Delta t} \right\}, \tag{28}$$

where $\boldsymbol{\epsilon}$ is the second-rank matrix with elements; $\epsilon_{11} = \epsilon^{(r)}$, $\epsilon_{22} = \epsilon^{(i)}$ and $\epsilon_{12} = \epsilon_{21} = 0$, while $[\cdots]^T$ denotes a vector transposition.

Using (28), we can write an explicit expression for the distribution:

$$P(u_1, u_2, t | u_1', u_2', t') = \frac{1}{2\pi \sqrt{\epsilon^{(r)} \epsilon^{(i)}} \Delta t} \times$$

$$\exp\left\{ -\frac{[u_1 - u_1' - k_1(u_1, u_2, t)\Delta t]^2}{2\epsilon^{(r)} \Delta t} - \frac{[u_2 - u_2' - k_2(u_1, u_2, t)\Delta t]^2}{2\epsilon^{(i)} \Delta t} \right\}, \tag{29}$$

where the coefficients $k_1$ and $k_2$ are defined from expression (15).

In the case when there is no dissipation in the environment, i.e., $\epsilon^{(i)} = 0$, the distribution (29) takes the following form:

$$P(u_1, u_2, t | u_1', u_2', t') = \frac{1}{2\pi \sqrt{\epsilon^{(r)} \Delta t}} \times$$

$$\exp\left\{ -\frac{[u_1 - u_1' - k_1(u_1, u_2, t)\Delta t]^2}{2\epsilon^{(r)} \Delta t} \right\} \delta[u_2 - u_2' + k_2(u_1, u_2, t)]. \tag{30}$$

As can be seen from the expression (29), the evolution of the system in the functional space $\mathbb{R}_{\{\xi\}}$ is characterized by a regular shift with a speed $\mathbf{K}(\mathbf{u}, t)$ against the background of Gaussian fluctuations with the diffusion matrix $\epsilon_{ij}$. Concerning to the trajectory $\mathbf{u}(t)$ in the space $\mathbb{R}_{\{\xi\}}$, it is determined by the following equations (see for example [20]):

$$\mathbf{u}(t) = \begin{cases} u_1(t + \Delta t) = u_1(t) + k_1(u_1, u_2, t)\Delta t + f^{(r)}(t)\Delta t^{1/2}, \\ u_2(t + \Delta t) = u_2(t) + k_2(u_1, u_2, t)\Delta t + f^{(i)}(t)\Delta t^{1/2}. \end{cases} \tag{31}$$

As can be seen from (31), the trajectory is continuous everywhere, i.e., $\mathbf{u}(t + \Delta t)\big|_{\Delta t \to 0} = \mathbf{u}(t)$, but it is nonetheless everywhere non-differentiable due to the presence of the term $\sim \Delta t^{1/2}$. If the time interval is represented as $\Delta t = t/N$, where $N \to \infty$, then expression (29) can be interpreted as the probability of transition from the vector $\mathbf{u}_l(t)$ to the vector $\mathbf{u}_{l+1}(t)$ during $\Delta t$ in the Brownian motion model.

Now, we can define the Fokker–Planck measure of the functional space:

$$D\mu(\mathbf{u}) = d\mu(\mathbf{u}_0) \lim_{N \to \infty} \left\{ \left( \frac{1}{2\pi} \frac{N/t}{\sqrt{\epsilon^{(r)} \epsilon^{(i)}}} \right)^N \prod_{l=0}^{N} du_{1(l+1)} du_{2(l+1)} \exp\left[ -\frac{N/t}{2\epsilon^{(r)}} \left( u_{1(l+1)} - \right. \right. \right.$$

$$\left. \left. \left. u_{1(l)} - k_{1(l+1)} \frac{t_{l+1}}{N} \right)^2 - \frac{N/t}{2\epsilon^{(i)}} \left( u_{2(l+1)} - u_{2(l)} - k_{2(l+1)} \frac{t_{l+1}}{N} \right)^2 \right] \right\}, \tag{32}$$

where $d\mu(\mathbf{u}_0) = \delta(u_1 - u_{1(0)})\delta(u_2 - u_{2(0)}) du_1 du_2$ denotes the measure of the initial distribution; in addition, the following notations are made:

$$u_{1(l)} = u_1(t_l), \quad u_{2(l)} = u_2(t_l), \quad k_{1(l)} = k_1(u_{1(l)}, u_{2(l)}, t_l), \quad k_{2(l)} = k_2(u_{1(l)}, u_{2(l)}, t_l).$$

Note that the measure $D\mu(\mathbf{w})$, which describes the probability of a given trajectory in the functional space $\mathbb{R}_{\{x\}}$, can be constructed in a similar way using the equation distributions for classical fields of the environment (25).

*3.2. The Stages of the Expected Trajectory Calculation*

We can now rigorously calculate the expected trajectory of the oscillator for the three cases described above.

**Definition 1.** *The functional integral along the random trajectory $\varrho\left[u_1(t), u_2(t), t\right]$ will be called the mathematical expectation of the trajectory:*

$$\bar{\varrho}(t) = \mathbb{E}[\varrho(t)] = \frac{1}{\alpha(t)} \int_{\mathbb{R}_{\{\xi\}}} D\mu(\mathbf{u})\varrho\left[u_1(t), u_2(t), t\right]. \tag{33}$$

*where $\alpha(t) = \int_{\mathbb{R}_{\{\xi\}}} D\mu(\mathbf{u}) = \int \int_{\Sigma_{\mathbf{u}}^{(2)}(t)} P(u_1, u_2, t)$ is a normalizing constant.*

Let us consider the case when the oscillator is not subjected by a random force, i.e., $F(t; \{\mathbf{g}\}) \equiv 0$. Then, the representation for the mathematical expectation of the trajectory, taking into account (8) and (33), will have the following form:

$$\bar{\xi}(t) = \mathbb{E}[\xi(t)] = \frac{\xi_0(t_0)}{\alpha(t)} \int_{\mathbb{R}_{\{\xi\}}} D\mu(\mathbf{u}) \exp\left\{\int_{t_0}^{t} \phi(t')dt'\right\}. \tag{34}$$

Computing the functional integral (34) using the generalized Feynman–Kac theorem [19], one can find the following two-dimensional integral representation for the mathematical expectation of the trajectory:

$$\bar{\xi}(t) = \mathbb{E}[\xi(t)] = \xi_0(t_0)\Lambda_Q(t), \qquad \Lambda_Q(t) = \frac{1}{\alpha(t)} \int \int_{\Sigma_{\mathbf{u}}^{(2)}(t)} Q(u_1, u_2, t)du_1 du_2, \tag{35}$$

where the function $Q(u_1, u_2, t)$ is the solution of the following second-order complex PDE:

$$\partial_t Q = \left\{\hat{L}(\mathbf{u}, t) + u_1 + iu_2\right\}Q. \tag{36}$$

Since $\xi_0(t_0)$ is a constant, the main role in determining the expectation of the trajectory is played by the function $\Lambda_Q(t)$.

A numerical study of a complex PDE (36) is carried out using the developed algorithm in the form of a system of difference equations (see **Listing 2** Section 7). The results of numerical simulation of the function $Q(u_1, u_2, t)$ depending on the state of the environment and time are shown in Figures 4–6 of Section 7.2. The mathematical expectation of the trajectory, depending on various parameters and time, was calculated and represented on the graphs (for details see Section 7.3, Figure 7).

Now, let us calculate the mathematical expectation of the trajectory $\bar{x}(t)$ when the oscillator is acted upon by a regular external force $F_0(t)$.

Given Equation (6), the trajectory can be formally written as follows:

$$\bar{x}(t) = \mathbb{E}\left[x(t)\right] = \frac{1}{\sqrt{2\Omega_0}}\left[I(t) + I^*(t)\right], \qquad I(t) = \left\langle\xi(t)d^*(t)\right\rangle, \tag{37}$$

where the symbol $\langle\cdots\rangle$ denotes functional integration with respect to the Fokker–Planck measure (see expression (32)):

$$I(t) = \left\langle\xi(t)d^*(t)\right\rangle = \int_{\mathbb{R}_{\{\xi\}}} D\mu(\mathbf{u})\xi(t)d^*(t). \tag{38}$$

The functional integral in (38) can be calculated and brought to a two-dimensional integral representation if we use the following auxiliary relation:

$$I(t) = I(m|t)\big|_{m=0} = \frac{i\bar{\xi}_0(t_0)}{\sqrt{2\Omega_0}} \partial_\lambda \left\langle \int_{t_0}^t \left[ \phi(t') + \lambda \xi(t') F_0(t') \right] dt' \right\rangle \bigg|_{\lambda=0}.$$

The value under the derivative sign can be calculated using the generalized Feynman–Kac theorem:

$$\left\langle \int_{t_0}^t \left[ \phi(t') + \lambda \xi(t') F_0(t') \right] dt' \right\rangle = \int\int_{\Sigma_{\mathbf{u}}^{(2)}(t)} Q_\lambda(u_1, u_2, t) du_1 du_2, \tag{39}$$

where the function $Q_\lambda(u_1, u_2, t)$ is the solution of the following second-order PDE:

$$\partial_t Q_\lambda = \left\{ \hat{L} + u_1 + iu_2 + \lambda \xi(t) F_0(t) \right\} Q_\lambda. \tag{40}$$

Differentiating Equation (40) with respect to the parameter $\lambda$, we find the following equation:

$$\partial_t \mathcal{D}_\lambda = \left\{ \hat{L} + u_1 + iu_2 + \lambda \xi(t) F_0(t) \right\} \mathcal{D}_\lambda + \xi(t) F_0(t) Q_\lambda,$$

where $\mathcal{D}_\lambda(u_1, u_2, t) \equiv \partial_\lambda Q_\lambda(u_1, u_2, t)$.

Now, introducing the notation $\mathcal{D}(u_1, u_2, t) = \mathcal{D}_\lambda(u_1, u_2, t)\big|_{\lambda=0}$, we obtain the following two-dimensional integral representation:

$$I(t) = \frac{i\bar{\xi}_0(t_0)}{\sqrt{2\Omega_0}} \Lambda_D(t), \qquad \Lambda_D(t) = \int\int_{\Sigma_{\mathbf{u}}^{(2)}(t)} \mathcal{D}(u_1, u_2, t) du_1 du_2,$$

where the function $\mathcal{D}(u_1, u_2, t)$ satisfies the following complex second-order PDE:

$$\partial_t \mathcal{D} = \left( \hat{L} + u_1 + iu_2 \right) \mathcal{D} + \xi(t) F_0(t) Q, \qquad Q \equiv Q_\lambda\big|_{\lambda=0}. \tag{41}$$

To solve Equation (40), it is necessary that the solution satisfies the following initial condition:

$$\mathcal{D}(u_1, u_2, t)\big|_{t=t_0} = 0. \tag{42}$$

As we can see, Equation (41) includes two external parameters; one of them is the regular function $F_0(t)$ (external force), and the other is the random oscillator trajectory $\xi(t) \in L_1$, which is a solution to stochastic Equation (12).

Integrating Equation (41) with respect to the Fokker–Planck measure (32), one can obtain a new equation:

$$\langle \partial_t \mathcal{D} \rangle = \left( \hat{L} + u_1 + iu_2 \right) \bar{\mathcal{D}} + \bar{\xi}(t) F_0(t) Q, \qquad \bar{\mathcal{D}} = \langle \mathcal{D} \rangle, \tag{43}$$

where $\bar{\xi}(t)$ denotes a regular function representing the mathematical expectation of the oscillator trajectory without external influence (see (34)). The $\langle \partial_t \mathcal{D} \rangle$ term in Equation (43) can be represented as:

$$\langle \partial_t \mathcal{D} \rangle = \partial_t \langle \mathcal{D} \rangle - \langle (\partial_t \mathcal{M}(t)) \mathcal{D} \rangle = \partial_t \bar{\mathcal{D}} - (\partial_t \mathcal{M}(t)) \bar{\mathcal{D}}, \tag{44}$$

where $\mathcal{M}(t)$ denotes the exponent of the Fokker–Planck measure (32) in the limit $N \to \infty$ when the sum goes into an integral.

It is easy to see that the term $\partial_t \mathcal{M}(t)$ is a random function and, accordingly, the new averaging does not allow one to obtain a regular equation for the distribution function $\bar{\mathcal{D}}$. However, since at large times $\partial_t \mathcal{D} \to 0$, then from Equation (43) in the asymptotic $t \to \infty$, we can obtain the following regular stationary equation:

$$\left( \hat{L} + u_1 + iu_2 \right) \bar{\mathcal{D}} + \bar{\xi}(t) F_0(t) Q = 0. \tag{45}$$

Taking into account (45), we can write the expectation of the oscillator trajectory in the external field when the process enters the asymptotic channel (*out*):

$$\bar{x}(t) = \frac{1}{\sqrt{2\Omega_0}}\{\bar{I}(t) + \bar{I}^*(t)\}, \qquad \bar{I}(t) = \frac{i\xi_0(t_0)}{\sqrt{2\Omega_0}} \int \int_{\Sigma_{\mathbf{u}}^{(2)}(t)} \bar{\mathcal{D}}(u_1, u_2, t) du_1 du_2. \qquad (46)$$

Considering that at $t \to \infty$, the ergodic properties of JS will prevail, we expect that the oscillator trajectory $\bar{x}(t)$ becomes a regular function of time.

In conclusion, we calculate the mathematical expectation of the oscillator trajectory under the action of an external random force $F(t; \{\mathbf{g}\})$. Carrying out similar reasoning, we can write the following functional integral for it:

$$\bar{x}_1(t) = \mathbb{E}\big[x_1(t)\big] = \frac{x_0(t_0)}{\alpha(t)} \int_{\mathbb{R}_{\{x\}}} D\mu(\mathbf{w}) \exp\left\{\int_{t_0}^{t} \theta(t')dt'\right\}. \qquad (47)$$

Doing a similar calculation in the functional integral (47), we obtain:

$$\bar{x}_1(t) = x_0(t_0)\Lambda_{Q_1}(t), \qquad \Lambda_{Q_1}(t) = \frac{1}{\alpha(t)} \int \int_{\Sigma_{\mathbf{w}}^{(2)}(t)} Q_1(w_1, w_2, t) dw_1 dw_2,$$

where $\Sigma_{\mathbf{w}}^{(2)}(t)$ is a two-dimensional manifold, the geometric and topological features of which must be studied specially. In addition, the function $x_0(t_0)$ is a solution to the regular Equation (23), and the function $Q_1(w_1, w_2, t)$ is a solution to the following complex PDE:

$$\partial_t Q_1 = \big\{\hat{\mathcal{L}}(\mathbf{w}, t) + w_1 + iw_2\big\}Q_1. \qquad (48)$$

As we can see, Equation (48) differs significantly from the parabolic complex PDE (36). It can go into the usual complex PDE in the $t \to \infty$ limit when JS comes to statistical equilibrium.

## 4. Geometric and Topological Features of a Compactified Space

As we saw in the previous section, in the limit of statistical equilibrium, the functional space $\mathbb{R}_{\{\xi\}}$ compactifies into the two-dimensional manifold. In particular, for a random frequency and no external force, the functional space $\mathbb{R}_{\{\xi\}}$ compactifies into the two-dimensional manifold $\Sigma_{\mathbf{u}}^{(2)}(t)$, and for a random external force and a regular frequency, the functional space, respectively, is compactified into another two-dimensional manifold; $\mathbb{R}_{\{\xi\}} \to \Sigma_{\mathbf{w}}^{(2)}(t)$. Thus, it is obvious that in this case, the JS in the limit of statistical equilibrium is described in three-dimensional space; $\mathbb{R}_{\bullet}^3 \cong \mathbb{R}^1 \otimes \Sigma_{\mathbf{u}}^{(2)}(t)$ or $\mathbb{R}_{\bullet}^3 \cong \mathbb{R}^1 \otimes \Sigma_{\mathbf{w}}^{(2)}(t)$, where $\mathbb{R}^1$ is a one-dimensional Euclidean subspace, and $\Sigma_{\mathbf{u}}^{(2)}(t)$ and $\Sigma_{\mathbf{w}}^{(2)}(t)$ are two-dimensional manifolds. Below, as an example, we will study in detail the topological and geometric features of the manifold $\Sigma_{\mathbf{u}}^{(2)}(t)$.

*4.1. Geometry of Two-Dimensional Subspace $\Sigma_{\mathbf{u}}^{(2)}(t)$*

**Definition 2.** *A generalized Riemannian (or pseudo-Riemannian) space is a smooth manifold $\Sigma_{\mathbf{u}}^{(2)}(t)$ with a doubly covariant tensor $g_{\mu\nu}$ defined on it, which we will call the generalized metric tensor.*

**Theorem 3.** *If the motion of a dynamical system is described by stochastic differential equations of the Langevin type (12), then in the limit of statistical equilibrium, these equations generate a two-dimensional space with an antisymmetric metric (Riemann–Cartan manifold).*

**Proof.** Let us represent Equations (18) and (19) in tensor form (see for example [21]):

$$\partial_t P = \nabla^2 P + k_0(u^1, u^2, t)P, \qquad \nabla^2 = \frac{1}{\sqrt{|g|}} \sum_{i,j=1}^{2} \frac{\partial}{\partial u^i} \left( \sqrt{|g|} g^{ij} \frac{\partial}{\partial u^j} \right), \tag{49}$$

where the following notations are made: $u_1 = u^1$ and $u_2 = u^2$.

To find the elements of the metric tensor, we write the two-dimensional Laplace-Beltrami operator $\nabla^2$ in explicit form:

$$\nabla^2 = g^{11} \frac{\partial^2}{\partial u_1^2} + \frac{1}{\sqrt{|g|}} \left[ \frac{\partial}{\partial u_1} \left( \sqrt{|g|} g^{11} \right) + \frac{\partial}{\partial u_2} \left( \sqrt{|g|} g^{21} \right) \right] \frac{\partial}{\partial u_1} + g^{12} \frac{\partial^2}{\partial u_1 \partial u_2}$$

$$+ g^{22} \frac{\partial^2}{\partial u_2^2} + \frac{1}{\sqrt{|g|}} \left[ \frac{\partial}{\partial u_2} \left( \sqrt{|g|} g^{22} \right) + \frac{\partial}{\partial u_1} \left( \sqrt{|g|} g^{12} \right) \right] \frac{\partial}{\partial u_2} + g^{21} \frac{\partial^2}{\partial u_2 \partial u_1}. \tag{50}$$

Comparing (50) with (19) and requiring the corresponding terms in the equations to be equal, we find:

$$g^{11} = \epsilon^{(r)}, \quad g^{22} = \epsilon^{(i)}, \quad g^{12} = -g^{21}, \quad g = g^{11}g^{22} - g^{12}g^{21} = \epsilon^{(r)}\epsilon^{(i)} + \left( g^{12} \right)^2. \tag{51}$$

As can be seen from (51), the metric tensor of the subspace $\Sigma_{\mathbf{u}}^{(2)}(t)$ is antisymmetric and, therefore, the corresponding geometry is non-commutative. Note that spaces with such properties arise both in mathematics and in quantum physics, and all of them are associated with non-commutative algebras [22]. For a deep understanding of non-commutative spaces—non-commutative versions of vector bundles, connections, curvature, etc.—as a rule, the operator algebra is used [23]. Recall that a non-commutative algebra is an associative algebra in which the multiplication is not commutative; i.e., $xy$ does not always equal $yx$. More generally, this algebraic structure, in which one of the basic binary operations is not commutative, furthermore allows for additional structures, such as topology or a norm, that can be carried over to a non-commutative functional algebra.

Before proceeding to the study of various properties of the $\Sigma_{\mathbf{u}}^{(2)}(t)$ subspace, we perform the following coordinate scaling transformation:

$$u_1 \to \bar{u}_1 = u_1 / \sqrt{\epsilon^{(r)}/\lambda}, \qquad u_2 \to \bar{u}_2 = u_2 / \sqrt{\epsilon^{(i)}/\lambda}, \tag{52}$$

where $\lambda > 0$ is some constant.

In this case, the metric tensor $\bar{g}^{ij}$ in the orthogonal basis can be represented as the following sum: $\bar{g}^{ij} = \lambda^{ij} + \bar{y}^{ij}$, $\lambda^{ij} = \lambda^{ji}$, $\bar{y}^{ij} = -\bar{y}^{ji}$), which can be represented explicitly:

$$\bar{g}^{ij} = \lambda \begin{pmatrix} 1 & 0 \\ 0 & 1 \end{pmatrix} + \bar{y} \begin{pmatrix} 0 & 1 \\ -1 & 0 \end{pmatrix}, \tag{53}$$

where $y(u_1, u_2, t) = g^{12}(u_1, u_2, t) \mapsto \bar{g}^{12}(\bar{u}_1, \bar{u}_2, t) = \bar{y}(\bar{u}_1, \bar{u}_2, t)$.

The first feature of the generalized metric is that the non-symmetric part does not contribute to the definition of the length, since $\bar{y}^{ij} v_i v_j = 0$, and therefore:

$$|\mathbf{v}| = \sqrt{\bar{g}^{ij} v_i v_j} = \sqrt{\lambda^{ij} v_i v_j + \bar{y}^{ij} v_i v_j} = \sqrt{\lambda^{ij} v_i v_j}.$$

Recall that the tensor $\lambda^{ij}$ defines the Euclidean geometry of the plane tangent to the manifold $\mathbf{v} \in \Sigma_{\mathbf{u}}^{(2)}(t)$ at a given point. In this symmetric space, $\lambda^{ij} = \lambda^{ji}$ angular measure and coordinates of the unit vector $\mathbf{v}$ are defined and equal to $\mathbf{v} = (\cos \vartheta, \sin \vartheta)$, respectively, where $\vartheta$ is the Euclidean angle between the vector $\mathbf{v}$ and the first basis vector.

Based on this definition and taking into account the antisymmetry property of the off-diagonal element of the metric tensor $\bar{g}^{12}(u_1, u_2, t) = -\bar{g}^{21}(u_1, u_2, t)$, it is easy to obtain two expressions for the cosine of an angle [24]:

$$\begin{cases} \cos(\widehat{\mathbf{v}_1, \mathbf{v}_2}) = \dfrac{\lambda^{ij}(v_1)_i(v_2)_j + \bar{y}^{ij}(v_1)_i(v_2)_j}{\sqrt{\lambda^{ij}(v_1)_i(v_1)_j}\sqrt{\lambda^{ij}(v_2)_i(v_2)_j}}, \\ \cos(\widehat{\mathbf{v}_2, \mathbf{v}_1}) = \dfrac{\lambda^{ij}(v_1)_i(v_2)_j - \bar{y}^{ij}(v_1)_i(v_2)_j}{\sqrt{\lambda^{ij}(v_1)_i(v_1)_j}\sqrt{\lambda^{ij}(v_2)_i(v_2)_j}}, \end{cases} \tag{54}$$

where $(v_1)_i$ denotes the projection of the $\mathbf{v}_1$ vector onto the $u_i$ axis (see (13)).

After doing some simple calculations, we find:

$$\begin{cases} \cos\psi^+ = \cos(\widehat{\mathbf{v}_1, \mathbf{v}_2}) = \sqrt{1 + (\bar{y}/\lambda)^2}\cos(\Delta\vartheta + \delta), \\ \cos\psi^- = \cos(\widehat{\mathbf{v}_2, \mathbf{v}_1}) = \sqrt{1 + (\bar{y}/\lambda)^2}\cos(\Delta\vartheta - \delta). \end{cases} \tag{55}$$

where $\Delta\vartheta = \vartheta_2 - \vartheta_1$ is the Euclidean angle between the vectors $\mathbf{v}_1$ and $\mathbf{v}_2$. As for the angle $\delta$, it is determined from the following relations:

$$\frac{\lambda}{\sqrt{\lambda^2 + \bar{y}^2}} = \cos\delta, \qquad \frac{\bar{y}}{\sqrt{\lambda^2 + \bar{y}^2}} = \sin\delta.$$

From (55) also follows the important conditions for the Euclidean angles $\vartheta^+ = \Delta\vartheta + \delta$ and $\vartheta^- = \Delta\vartheta - \delta$. In particular, it follows from the definition of Euclidean geometry that the angles must satisfy the following constraint conditions:

$$\cos\vartheta^+ \leq \frac{1}{\sqrt{1 + (\bar{y}/\lambda)^2}}, \qquad \cos\vartheta^- \leq \frac{1}{\sqrt{1 - (\bar{y}/\lambda)^2}}. \tag{56}$$

Recall that two different values of the angle $\psi^+$ and $\psi^-$ between the vectors $\mathbf{v}_1 = \mathbf{v}_1(u_1, u_2)$ and $\mathbf{v}_2 = \mathbf{v}_2(u_1, u_2)$ (see (55)) depending on the direction of rotation—to the right or to the left— is a characteristic peculiarity of Kozyrev's theory [25].

Taking into account the antisymmetry of the metric of the two-dimensional space $\Sigma_{\mathbf{u}}^{(2)}(t)$, it is easy to prove that its Gaussian curvature is equal to zero. However, following Cartan [26,27], one can introduce a generalized linear connection:

$$G_{jk}^i = \Gamma_{jk}^i + K_{jk}^i, \qquad i, j, k = 1, 2, \tag{57}$$

where $\Gamma_{jk}^i = \frac{1}{2}\lambda^{il}(\lambda_{lj;k} + \lambda_{lk;j} - \lambda_{jk;l})$, $(\lambda_{lj;k} = \partial\lambda_{lj}/\partial\bar{u}_k)$ is the Christoffel symbol and $K_{jk}^i$ denotes the curvature tensor formed by the interaction of an oscillator with a random medium. This tensor can be defined as follows:

$$K_{ijk}(\bar{u}_1, \bar{u}_2, t; \bar{y}) = \frac{1}{2}\left(\frac{\partial w_i}{\partial\bar{u}^k} - \frac{\partial w_k}{\partial\bar{u}^i}\right)\bar{u}_j, \qquad w_i = \bar{y}\delta_{ii},$$

where $\delta_{ij}$ denotes the Kronecker symbol.

Note that since the tensor $K_{ijk}(\bar{u}_1, \bar{u}_2, t; \bar{y})$ is antisymmetric with respect to the first pair of indices, then the connection $G_{ijk}(\bar{u}_1, \bar{u}_2, t; \bar{y})$ is consistent with the metric.

Now, we can write the equation of motion of a quasi-particle or excitation of an environment, which, taking into account the representation (53) will have the following form:

$$\frac{\partial^2\bar{u}^i}{\partial s^2} + K_{jk}^i(\bar{u}_1, \bar{u}_2, t; \bar{y})\dot{\bar{u}}^j\dot{\bar{u}}^k = 0, \qquad \Gamma_{jk}^i \equiv 0, \quad i, j, k = 1, 2, \tag{58}$$

where $s = \int\sqrt{\lambda^{ij}du_i du_j} = \int\sqrt{d\bar{u}_i d\bar{u}^i}$ and $\dot{\bar{u}}^i = \partial\bar{u}^i/\partial s$.

Thus, Equation (58) can be considered as the equation of a geodesic line in space with connection $K^i_{jk}(\bar{u}_1, \bar{u}_2, t; \bar{y})$. To solve this equation, it is necessary to know the form of the contortion tensor $K^i_{jk}(\bar{u}_1, \bar{u}_2, t; \bar{y})$ as a function of coordinates and time. □

*4.2. Topology of Two-Dimensional Subspace $\Sigma^{(2)}_{\mathbf{u}}(t)$*

**Theorem 4.** *The generalized metric $g^{ij}(u_1, u_2, t)$ satisfying (49)–(51) is defined by the antisymmetric metric element $g^{12}(u_1, u_2, t) = -g^{21}(u_1, u_2, t)$, which satisfies a fourth-degree algebraic equation generating a topological manifold with possibly non-trivial singularities and a first Betti number less than or equal to 4.*

**Proof.** Comparing the operators (19) and (50) and taking into account the coordinate transformations (52), we can obtain the following first-order partial differential equations for the antisymmetric element of the metric tensor $y = g^{12}(u_1, u_2, t)$:

$$\varepsilon^{(r)}\chi\frac{\partial y}{\partial u_1} - (1 + y\chi)\frac{\partial y}{\partial u_2} = k_1(u_1, u_2, t),$$

$$\varepsilon^{(i)}\chi\frac{\partial y}{\partial u_2} + (1 + y\chi)\frac{\partial y}{\partial u_1} = k_2(u_1, u_2, t), \tag{59}$$

where $\chi = y/(a + y^2)$ and $a = \varepsilon^{(r)}\varepsilon^{(i)}$.

However, as follows from (59), the system of equations is redefined with respect to the desired function $y$. In this regard, a reasonable question arises: under what conditions are these equations compatible?

From Equation (59), it is easy to find expressions for two different derivatives of the off-diagonal component of the metric tensor:

$$y_1 = \frac{\partial y}{\partial u_1} = \frac{\varepsilon^{(i)}k_1\chi + k_2(1 + y\chi)}{a\chi^2 + (1 + y\chi)^2}, \qquad y_2 = \frac{\partial y}{\partial u_2} = \frac{\varepsilon^{(r)}k_2\chi - k_1(1 + y\chi)}{a\chi^2 + (1 + y\chi)^2}. \tag{60}$$

Using Equation (60), we can find the following two expressions for the mixed second derivatives of the metric tensor element $y = g^{12}(u_1, u_2, t)$:

$$y_{12} = \frac{\partial^2 y}{\partial u_2 \partial u_1} = \frac{\varepsilon^{(i)}(k_{1;2}\chi + k_1\chi_2) + k_{2;2}(1 + y\chi) + k_2(y_2\chi + y\chi_{;2})}{a\chi^2 + (1 + y\chi)^2}$$
$$- 2\frac{\varepsilon^{(i)}k_1\chi + k_2(1 + y\chi)}{[a\chi^2 + (1 + y\chi)^2]^2}[a\chi\chi_{;2} + (1 + y\chi)(y_2\chi + y\chi_{;2})],$$

$$y_{21} = \frac{\partial^2 y}{\partial u_1 \partial u_2} = \frac{\varepsilon^{(r)}(k_{2;1}\chi + k_2\chi_{;1}) - k_{1;1}(1 + y\chi) - k_1(y_1\chi + y\chi_{;1})}{a\chi^2 + (1 + y\chi)^2}$$
$$- 2\frac{\varepsilon^{(r)}k_2\chi - k_1(1 + y\chi)}{[a\chi^2 + (1 + y\chi)^2]^2}[a\chi\chi_{;2} + (1 + y\chi)(y_1\chi + y\chi_{;1})], \tag{61}$$

where $\chi_{;j} = \partial\chi/\partial u_j$ and $k_{i;j} = \partial k_i/\partial u_j$, $(i, j = 1, 2)$.

It is important to note that the antisymmetry of the off-diagonal elements of the metric tensor arises at the stage of choosing the coordinate system and, accordingly, the orientation of the considered sub-manifold.

As for the question of the symmetry of mixed second derivatives on any oriented manifolds, then, based on the basic requirement of mathematical analysis, the following identity must hold at any time at each point of the manifold (Schwarz's theorem, see [28]):

$$y_{12} = \frac{\partial^2 y}{\partial u_1 \partial u_2} = y_{21} = \frac{\partial^2 y}{\partial u_2 \partial u_1}, \tag{62}$$

which is a necessary condition for a twice continuously differentiable function. In the context of partial differential equations, it is called the Schwarz integrability condition.

If we write this equality (62) explicitly, taking into account expressions (61), then it will look like this:

$$2\left[\varepsilon^{(r)}k_2\chi - k_1(1+y\chi)\right]\frac{a\chi\chi_{;1} + (1+y\chi)(y_1\chi + y\chi_1)}{a\chi^2 + (1+y\chi)^2} + (k_1 + k_2 y)\chi_{;2}$$

$$= 2\left[\varepsilon^{(i)}k_1\chi + k_2(1+y\chi)\right]\frac{a\chi\chi_{;2} + (1+y\chi)(y_2\chi + y\chi_{;2})}{a\chi^2 + (1+y\chi)^2} + (k_2 - k_1 y)\chi_{;1}$$

$$- (k_{1;1} + k_{2;2})(1+y\chi) - (k_1 y_1 + k_2 y_2 + \varepsilon^{(i)}k_{1;2} - \varepsilon^{(r)}k_{2;1})\chi. \tag{63}$$

Finally, given (60) from Equation (63), we obtain the following 4th degree algebraic equation:

$$\sum_{n=0}^{4} A_n(u_1, u_2, t)y^n = 0, \tag{64}$$

where the coefficients of the algebraic equation $A_n(u_1, u_2, t)$ are defined by the expressions:

$$A_0 = a\left\{4au_1 - 4\varepsilon^{(r)}u_1^2 u_2^2 - \varepsilon^{(i)}[u_1^2 - u_2^2 + \Omega_0^2(t)]^2\right\}, \quad A_1 = -2au_2(\varepsilon^{(r)} + \varepsilon^{(i)}), \quad A_2 =$$

$$24au_1 + 8\varepsilon^{(r)}u_1^2 u_2^2 + 2\varepsilon^{(i)}[u_1^2 - u_2^2 + \Omega_0^2(t)]^2, \quad A_3 = -8u_2(\varepsilon^{(r)} + \varepsilon^{(i)}), \quad A_4 = 32u_1.$$

The general 4th degree Equation (64) is solved exactly by the Ferrari method and has four solutions, some of which may be complex [29]. Since the coefficients of the Equation (64) are functions of two coordinates $(u_1, u_2)$ and time, the solutions must form a continuum of sets in two-dimensional Euclidean space. We will be interested in those sets of solutions that are complex. Obviously, if we cut out and remove from the Euclidean space all the domains on which the solutions of the algebraic Equation (64) are complex, then the remaining space will have topological singularities. As the numerical solution of the algebraic Equation (64) shows, depending on the parameters $\varepsilon^{(r)}$ and $\varepsilon^{(i)}$, the manifold $\Sigma_{\mathbf{u}}^{(2)}(t)$ has the first Betti number $n \leq 4$, where 4 is the number of complex solutions of this equation.

Thus, we have proved the necessary condition for the compatibility of the system of Equation (59). To prove whether this condition is sufficient for the compatibility of the system of Equation (64), it is necessary to substitute the solution of the algebraic Equation (64) into this system. Unfortunately, this assertion verification procedure turns out to be a very difficult task. However, due to the fact that all sequential mathematical constructions are performed analytically by algebraic methods, there is every reason to believe that inverse transformations also take place; i.e., up to linear functions $u_1$ and $u_2$, the function $y$ is a solution to the Equation (59). □

To illustrate the above, below are the results of visualization of a series of calculations (see Figures 8–11), which allow obtaining a detailed idea of the topological features of the two-dimensional manifold $\Sigma_{\mathbf{u}}^{(2)}(t)$, which arises after the compactification of the function space $\mathbb{R}_{\{x\}}$ (see Section 7.4 of Section 7 for details). It is also important to note that a similar analysis for the complex Equation (36) describing the solution $Q(u_1, u_2, t)$ proves that the exact geometry for solving this problem is the manifold $\Sigma_{\mathbf{u}}^{(2)}(t)$.

## 5. Statement of the Initial-Boundary Value Problem for the Complex PDE

For definiteness, we will study the initial-boundary value problem in the case when an external force does not act on the oscillator in the thermostat. In this case, the mathematical expectation of the trajectory is described by a two-dimensional integral representation (35),

where the function $Q(u_1, u_2, t)$ is the solution of a complex second-order PDE (36). We will consider the problem when $\Sigma_{\mathbf{u}}^{(2)}(t)$ is a two-dimensional Euclidean space, that is:

$$\Sigma_{\mathbf{u}}^{(2)}(t) \cong \mathbb{R}^2 \equiv (-\infty, +\infty) \times (-\infty, +\infty)$$

**Theorem 5.** *If a two-dimensional complex PDE has the form (36), then using the internal symmetry of the equation, it can be reduced to two independent PDFs belonging to the class of PDFs with a deviated argument given by affine transformations such as reflection.*

**Proof.** Representing the solution of Equation (36) as a sum of real and imaginary parts:

$$Q(u_1, u_2, t) = Q^{(r)}(u_1, u_2, t) + iQ^{(i)}(u_1, u_2, t), \tag{65}$$

one can obtain the following system of PDEs:

$$\begin{cases} \partial_t Q^{(i)}(u_1, u_2, t) = \{\hat{L} + u_1\}Q^{(i)}(u_1, u_2, t) + u_2 Q^{(r)}(u_1, u_2, t), \\ \partial_t Q^{(r)}(u_1, u_2, t) = \{\hat{L} + u_1\}Q^{(r)}(u_1, u_2, t) - u_2 Q^{(i)}(u_1, u_2, t). \end{cases} \tag{66}$$

The functions $Q^{(i)}(u_1, u_2, t)$ and $Q^{(r)}(u_1, u_2, t)$ can be normalized and given the meaning of the probability density:

$$\bar{Q}^{(i)}(u_1, u_2, t) = \beta^{-1}(t)Q^{(i)}(u_1, u_2, t), \qquad \bar{Q}^{(r)}(u_1, u_2, t) = \beta^{-1}(t)Q^{(r)}(u_1, u_2, t),$$

$$\beta(t) = \int\int_{\Sigma_{\mathbf{u}}^{(2)}(t)} \left[Q^{(i)}(u_1, u_2, t) + Q^{(r)}(u_1, u_2, t)\right] du_1 du_2. \tag{67}$$

and, obviously, the functions must satisfy the normalization condition:

$$\int\int_{\Sigma_{\mathbf{u}}^{(2)}(t)} \left[\bar{Q}^{(i)}(u_1, u_2, t) + \bar{Q}^{(r)}(u_1, u_2, t)\right] du_1 du_2 = 1.$$

It is easy to see that when changing the coordinates $(u_1, u_2) \to (u_1, -u_2)$, the system of Equation (66) becomes:

$$\begin{cases} \partial_t Q^{(i)}(u_1, -u_2, t) = \{\hat{L} + u_1\}Q^{(i)}(u_1, -u_2, t) - u_2 Q^{(r)}(u_1, -u_2, t), \\ \partial_t Q^{(r)}(u_1, -u_2, t) = \{\hat{L} + u_1\}Q^{(r)}(u_1, -u_2, t) + u_2 Q^{(i)}(u_1, -u_2, t). \end{cases} \tag{68}$$

If we assume that $Q^{(i)}(u_1, -u_2, t) = Q^{(r)}(u_1, u_2, t)$ and, accordingly, $Q^{(r)}(u_1, -u_2, t) = Q^{(i)}(u_1, u_2, t)$, then the system of Equation (68) takes the original form (66), i.e., the first equation goes over into the second, and the second goes into the first. Using these obvious symmetry properties, we can write the system of coupled PDEs (68) as two independent PDEs:

$$\begin{cases} \partial_t Q^{(i)}(u_1, u_2, t) = \{\hat{L} + u_1\}Q^{(i)}(u_1, u_2, t) - u_2 Q^{(i)}(u_1, -u_2, t), \\ \partial_t Q^{(r)}(u_1, u_2, t) = \{\hat{L} + u_1\}Q^{(r)}(u_1, u_2, t) + u_2 Q^{(r)}(u_1, -u_2, t). \end{cases} \tag{69}$$

As we can see in the system (69), the equations are independent, and each of them belongs to the PDE class with a spatially deviated argument given by affine transformations such as reflection. By solving one of the equations, we can obtain the solution of the second one using a 180-degree rotation in the two-dimensional Euclidean space $\mathbb{R}^2$. Before proceeding to the numerical solution of these PDEs, we consider three possible scenarios:

(a) When the solutions of Equation (69) are even functions with respect to the $u_2$ coordinate, i.e., $Q^{(i)}(u_1, -u_2, t) = Q^{(i)}(u_1, u_2, t)$ and $Q^{(r)}(u_1, -u_2, t) = Q^{(r)}(u_1, u_2, t)$;

(b) When the solutions of Equation (69) are odd functions with respect to the coordinate $u_2$, i.e., $Q^{(i)}(u_1, -u_2, t) = -Q^{(i)}(u_1, u_2, t)$ and $Q^{(r)}(u_1, -u_2, t) = -Q^{(r)}(u_1, u_2, t)$, and, accordingly, the case;

(c) When the indicated functions do not have definite parity.

In the first (a) case from (69), we obtain two unrelated PDEs:

$$
\begin{cases}
\partial_t Q^{(i)}(u_1, u_2, t) = \{\hat{L} + u_1\} Q^{(i)}(u_1, u_2, t) - u_2 Q^{(i)}(u_1, u_2, t), \\
\partial_t Q^{(r)}(u_1, u_2, t) = \{\hat{L} + u_1\} Q^{(r)}(u_1, u_2, t) + u_2 Q^{(r)}(u_1, u_2, t).
\end{cases}
\tag{70}
$$

In the second (b) case, we again obtain a system of uncoupled differential equations, only in this case, it is necessary to replace $Q^{(i)}(u_1, u_2, t) \to Q^{(r)}(u_1, u_2, t)$ in the first equation and $Q^{(r)}(u_1, u_2, t) \to Q^{(i)}(u_1, u_2, t)$ in the second one, respectively.

In the third (c) case, the functions $Q^{(i)}(u_1, u_2, t)$ and $Q^{(r)}(u_1, u_2, t)$ are described by the system of Equation (69). Note that this is the most general and difficult case for numerical simulation, which will be considered in detail below. $\square$

We will consider a more complicated case where the PDE is a deviant argument. Our task will be to formulate an initial-boundary value problem for solving one of the PDEs of the equations system (69).

For definiteness, let us consider the second equation in (69). We can represent it as the following system:

$$
\begin{cases}
\partial_t\, Q^{(r)}(u_1, u_2, t) = \{\hat{L} + u_1\} Q^{(r)}(u_1, u_2, t) + u_2 Q^{(r)}(u_1, -u_2, t), \\
\partial_t Q^{(r)}(u_1, -u_2, t) = \{\hat{L} + u_1\} Q^{(r)}(u_1, -u_2, t) + u_2 Q^{(r)}(u_1, u_2, t).
\end{cases}
\tag{71}
$$

Recall that the second equation in (71) is obtained from the first one by replacing $u_2 \to -u_2$.

As an initial condition, we assume that the probability distribution of the environmental fields $Q^{(r)}(u_1, u_2, t)$ at time $t_0$ is described by the Dirac delta function:

$$
Q^{(r)}(u_1, u_2, t_0) = \prod_{j=1}^{2} \delta(u_j - u_{0j}), \qquad u_{0j} = u_j(t_0).
\tag{72}
$$

As for the boundary conditions, we define the Neumann boundary conditions on the perpendicular axes $u_1$ and $u_2$, respectively:

$$
\frac{\partial}{\partial u_2} Q^{(r)}(u_1, u_2, t)\Big|_{u_2=0} = Q^{(r)}_{;2}(u_1, 0, t), \qquad \frac{\partial}{\partial u_1} Q^{(r)}(u_1, u_2, t)\Big|_{u_1=0} = Q^{(r)}_{;1}(0, u_2, t), \tag{73}
$$

where $Q^{(r)}_{;i} = \partial Q^{(r)} / \partial u_i, \quad (i = 1, 2)$. We will determine their values based on a number of physical considerations. As we will see below, the conditions (73) lead to two different equations.

First, we consider the behavior of the second equation in (69) near the $u_1 \in (-\infty, +\infty)$ axis. The solution of the equation near this axis can be represented as:

$$
Q^{(r)}(u_1, u_2, t)\Big|_{u_2 \sim 0} = (a_0 + a_1 u_2 + a_2 u_2^2 + \cdots) e^{-u_2^2/2} \mathbb{Q}_1^{(r)}(u_1, t),
\tag{74}
$$

where $a_0, a_1$ and $a_2$ are some unknown constants that will be determined based on physical considerations.

Substituting (74) into the second equation in (69), in the limit of $u_2 \to 0$, one obtains the following second-order PDE:

$$
\frac{\partial}{\partial t} \mathbb{Q}_1^{(r)} = \left\{ \epsilon^{(r)} \frac{\partial^2}{\partial u_1^2} + \left(u_1^2 + \Omega_0^2(t)\right) \frac{\partial}{\partial u_1} + \left[ 5u_1 - \epsilon^{(i)} \left( \frac{2a_2}{a_0} - 1 \right) \right] \right\} \mathbb{Q}_1^{(r)}.
\tag{75}
$$

In particular, for the first Neumann condition, we obtain:

$$
\frac{\partial}{\partial u_2} Q^{(r)}(u_1, u_2, t)\Big|_{u_2=0} = Q^{(r)}_{;2}(u_1, 0, t) = a_1 \mathbb{Q}_1^{(r)}(u_1, t).
\tag{76}
$$

Since the equation for the function $Q^{(r)}(u_1, u_2, t)$ is not symmetric with respect to the change $u_2 \rightarrow -u_2$, the first Neumann boundary condition cannot be equal to zero and, accordingly, $a_1 \neq 0$. In addition, the function $Q_{;2}^{(r)}(u_1, t) \sim \mathbb{Q}_1^{(r)}(u_1, t)$ has the meaning of the probability density on the $u_1$ axis; we can normalize it to one. The latter means that $a_1 = 1$ and the normalization of the solution $\mathbb{Q}_1^{(r)}(u_1, t)$ is equal to one, that is:

$$\bar{\mathbb{Q}}_1^{(r)}(u_1, t) = c_0^{-1}(t)\mathbb{Q}_1^{(r)}(u_1, t), \qquad c_0(t) = \int_{-\infty}^{+\infty} \mathbb{Q}_1^{(r)}(u_1, t)du_1,$$

where $c_0(t)$ is the normalization constant.

Proceeding from the fact that the boundary conditions on the perpendicular axes $u_1$ and $u_2$ describe the probability distributions of elastic and inelastic collisions independent of each other, it is natural to assume that the term of inelastic collision in Equation (75) should be identically equal to zero. In other words, we can require that the equality $2a_2/a_0 - 1 = 0$ becomes satisfied, and thus, Equation (75) can be written as:

$$\frac{\partial}{\partial t}\mathbb{Q}_1^{(r)} = \left\{ \epsilon^{(r)}\frac{\partial^2}{\partial u_1^2} + (u_1^2 + \Omega_0^2(t))\frac{\partial}{\partial u_1} + 5u_1 \right\}\mathbb{Q}_1^{(r)}. \tag{77}$$

To solve this equation, it is necessary to formulate the following initial-boundary value problem:

$$\mathbb{Q}_1^{(r)}(u_1, t)\big|_{t=t_0} = \delta(u_1 - u_{01}), \tag{78}$$

and, correspondingly,

$$\mathbb{Q}_1^{(r)}(u_1, t)\big|_{u_1=s_1} = 0, \qquad \mathbb{Q}_1^{(r)}(u_1, t)\big|_{u_1=s_2} = 0. \tag{79}$$

where $s_1$ and $s_2$ denote points far enough from the origin 0, which are located to the left and right, respectively.

Now, to establish the second Neumann boundary condition, we consider the solution $Q^{(r)}(u_1, u_2, t)$ near the axis $u_2 \in (-\infty, +\infty)$. To do this, we represent the solution in the form:

$$Q^{(r)}(u_1, u_2, t)\big|_{u_1 \sim 0} = (b_0 + b_1 u_1 + b_2 u_1^2 + \cdots)e^{-u_1^2/2}\mathbb{Q}_2^{(r)}(u_2, t), \tag{80}$$

where $b_0$, $b_1$ and $b_2$ are some constants that we will define below.

Substituting (80) into (69), in the limit of $u_1 \rightarrow 0$, we obtain the following equation:

$$\frac{\partial}{\partial t}\mathbb{Q}_2^{(r)}(u_2, t) =$$
$$\left\{ \epsilon^{(i)}\frac{\partial^2}{\partial u_2^2} + \left[ \epsilon^{(r)}\left(\frac{2b_2}{b_0} - 1\right) + \frac{b_1}{b_0}\left(\Omega_0^2(t) - u_2^2\right) \right] \right\}\mathbb{Q}_2^{(r)}(u_2, t) + u_2\mathbb{Q}_2^{(r)}(-u_2, t). \tag{81}$$

Since only inelastic processes are taken into account on the $u_2$ axis, we can require that the following condition becomes fulfilled: $2b_2/b_0 - 1 = 0$. In addition, for definiteness, we can set the constant $b_0 = 1$ and $b_1 = 1/2$. Taking into account the clarifications made, Equation (82) can be simplified and presented as:

$$\frac{\partial}{\partial t}\mathbb{Q}_2^{(r)}(u_2, t) = \left\{ \epsilon^{(i)}\frac{\partial^2}{\partial u_2^2} + \frac{1}{2}\left(\Omega_0^2(t) - u_2^2\right) \right\}\mathbb{Q}_2^{(r)}(u_2, t) + u_2\mathbb{Q}_2^{(r)}(-u_2, t). \tag{82}$$

Because (82) is a second-order PDE with a deviant argument, it can be solved together with the same equation but after replacing the argument $u_2 \rightarrow -u_2$. For Equation (82), we can formulate the following initial-boundary conditions:

$$\mathbb{Q}_2^{(r)}(u_2, t)\big|_{t=t_0} = \delta(u_2 - u_{02}),$$

and, correspondingly,

$$\mathbb{Q}_2^{(r)}(u_2, t)\big|_{u_2 = \pm s} = 0, \qquad |s| \gg 1.$$

Considering the above, for the second boundary condition, we can write the following expression:

$$\frac{\partial}{\partial u_1} Q^{(r)}(u_1, u_2, t)\big|_{u_1 = 0} = Q_{2;1}^{(r)}(0, u_2, t) = b_1 \mathbb{Q}_2^{(r)}(u_2, t) = \frac{1}{2}\mathbb{Q}_2^{(r)}(u_2, t). \tag{83}$$

As in the case of the solution $\mathbb{Q}_1^{(r)}(u_1, t)$, we can interpret the solution $\mathbb{Q}_2^{(r)}(u_2, to)$ as the density probability and normalize it to one.

## 6. Entropy of a Self-Organizing System

As known, for a classical dynamical system, an important characteristic is the non-stationary Shannon entropy [30]. In particular, the entropy production rate is a quantitative measure of a non-equilibrium processes, and knowledge of its quantity indicates information about the dissipated heat [31,32], the difference in free energy between two equilibrium states [33,34], and also about the efficiency, if the considered non-equilibrium system is an engine [35–37]. It should be noted that the rate of entropy production provides important information for systems with hidden degrees of freedom [38,39], as well as for interacting subsystems, where the amount of information plays a key role [40–43].

In philosophy, physics and mathematics, the term negentropy is often used, which has a negative value and, therefore, has the opposite to entropy sense. Note that if entropy characterizes the measure of orderliness and organization of the system, then negentropy is the possibility of reducing entropy or an effort toward order. Recall that this concept was first proposed by Schrödinger [44] when explaining the behavior of living systems: *in order not to die, a living system struggles with the surrounding chaos and with the entropy it produces, organizing and ordering the latter by introducing negentropy*. This, in particular, explains the behavior of self-organizing systems.

For definiteness, let us consider the question of the change in the entropy of a classical oscillator in the case when its frequency has a random component. We first calculate the dynamics of the oscillator entropy without taking into account its influence on the random environment, when, as in the second case, we will consider this influence consistently and strictly. In particular, in the first case, we can define the non-stationary entropy in the standard way:

$$\mathcal{S}(t) = -\int\int_{\Sigma_{\mathbf{u}}^{(2)}(t)} \bar{P}(u_1, u_2, t) \ln \bar{P}(u_1, u_2, t) du_1 du_2. \tag{84}$$

It is often important to know the change in entropy or entropy production over a certain period of time:

$$\Delta\mathcal{S}(t_1, t_2) \equiv \mathcal{S}(t_2) - \mathcal{S}(t_1).$$

Since processes of different nature occur in the problem under consideration, we can also introduce the concept of partial entropy, which characterizes the production of entropy of a particular process:

$$\mathcal{S}_{par}^{(\sigma)}(t) = -\int\int_{\Sigma_{\mathbf{u}}^{(2)}(t)} \bar{Q}^{(\sigma)}(u_1, u_2, t) \ln \bar{Q}^{(\sigma)}(u_1, u_2, t) du_1 du_2, \qquad \sigma = i, r. \tag{85}$$

Note that the partial entropies $\mathcal{S}_{par}^{(r)}(t)$ and $\mathcal{S}_{par}^{(i)}(t)$ are in any case related by the total probability normalization condition (67) and, accordingly, will influence each other in the course of evolution. Recall that the partial entropy $\mathcal{S}_{par}^{(r)}(t)$ characterizes the processes of elastic collisions of the oscillator with the environment, while $\mathcal{S}_{par}^{(i)}(t)$ describes the processes of inelastic collisions of the oscillator with the environment. The study of partial

entropies will obviously provide additional important information about a dynamical system immersed in a thermostat.

Finally, we can determine the entropy of a closed self-organizing system classical oscillator + thermostat. In this case, the expression for the generalized Shannon entropy $\mathcal{S}_{gen}(t)$ can be represented as:

$$\mathcal{S}_{gen}(t) = - \int \int_{\Sigma_{\mathbf{u}}^{(2)}(t)} \Big( \sum_{\sigma=i,r} \bar{Q}^{(\sigma)}(u_1, u_2, t) \Big) \ln \Big( \sum_{\sigma=i,r} \bar{Q}^{(\sigma)}(u_1, u_2, t) \Big) du_1 du_2. \tag{86}$$

As can be seen from the graphs (see Figure 11), at intermediate times, the behavior of the entropies calculated by the formulas (84) and (86) differ greatly in value and in character. Moreover, in some time intervals, the generalized entropy (86), regardless of the intensity of the processes in the environment, takes a negative value, which is quite natural for a self-organizing system. Finally, as follows from the simulation and the corresponding visualizations, the behavior of both ordinary and generalized entropy tends to a constant value in the limit of large times.

Details of entropy calculations are discussed in Sub-section **E** of Section 7.

## 7. Numerical Methods for Solving the Problem

Numerical simulation of the self-organization processes in the joint system (classical oscillator and thermostat), even for a simple case, i.e., in the absence of an external field $F(t; \{\mathbf{g}\}) \equiv 0$, requires large computational resources. This is due to the fact that the complex probabilistic processes described by the three key PDEs, Equations (18) and (19), as well as the system of Equation (66), are numerically difficult solvable tasks. Recall that in [9], we have already considered a PDE of this type. In the same work, we considered various finite-difference methods of solution. Taking into account the analysis performed and test calculations for the numerical solution, we chose an explicit finite-difference scheme of the second order of accuracy in coordinates and the first order in time. Despite the external simplicity of this method, we believe that this scheme satisfies the goals of our work in terms of efficiency and accuracy. Note that Equations (18), (19) and (66) are second-order PDEs of the parabolic type with a source term. The main difficulty that we face in the calculations arises in connection with the convective transfer, which must be taken into account when increasing the size of the computational domain.

Below, we consider a numerical algorithm for solving the initial-boundary value problems for these PDEs in the two-dimensional Euclidean space $\mathbb{R}^2$, as shown in Listings 1 and 2.

For a PDE system (88) with appropriate conditions on the coordinate axes, the finite difference scheme is similar to Listing 1.

In view of the symmetry of Equations (87) and (88) with respect to the coordinate $\bar{u}_2$, the calculation was carried out for the upper half-plane, i.e., for the $\bar{u}_2 \geq 0$ for a square grid of $600 \times 400$ nodes in $\bar{u}_1 \times \bar{u}_2$, respectively. Then, the results were recalculated for the entire region $\mathbb{R}^2$, that is on the $600 \times 800$ grid for all solutions $P(\bar{u}_1, \bar{u}_2, t)$, $Q^{(r)}(\bar{u}_1, \bar{u}_2, t)$ and $Q^{(i)}(\bar{u}_1, \bar{u}_2, t)$. Space steps, $\Delta \bar{u}_1 = \Delta \bar{u}_2 = 0.02$, time step for the first option, $\Delta t = 10^{-5}$; for the second and third options, $\Delta t = 2 \times 10^{-5}$.

Note that in this work, the proposed difference scheme for a linear two-dimensional equation with a source term, which can be attributed to the parabolization schemes for equations of the convection–diffusion type, has no independent meaning. The stability conditions for such schemes have been considered in sufficient detail in various monographs; see, for example, [45]. Let us present the necessary stability conditions for scheme (87). The conditions for scheme (88) look similar. Let us rewrite expression (87) in the following form, ignoring the source term $4 \Delta t P_{j,k}^n$:

$$P_{j,k}^{n+1} = P_{j,k}^n + r_1 \big[ P_{j+1,k}^n - 2 P_{j,k}^n + P_{j-1,k}^n \big] + r_2 \big[ P_{j,k+1}^n - 2 P_{j,k}^n + P_{j,k-1}^n \big] +$$

$$C_1 \big[ P_{j+1,k}^n - P_{j-1,k}^n \big] + C_2 \big[ P_{1,k+1}^n - P_{j,k-1}^n \big],$$

where

$$C_1 = \frac{\Delta t}{2\Delta \bar{u}_1}\left[(\bar{u}_1^2)_j - (\bar{u}_2^2)_k - \Omega_{0(n)}\right], \qquad C_2 = \frac{\Delta t}{\Delta \bar{u}_2}(\bar{u}_1)_j(\bar{u}_2)_k.$$

**Listing 1.** Numerical algorithm for solving PDEs (18) and (19) after coordinate transformations (52).

1. The continuous area $\mathbb{R}^2$ for Equations (18) and (19) is replaced by the calculated area: $[(\bar{u}_1)_{\min}, (\bar{u}_1)_{\max}] \times [(\bar{u}_2)_{\min}, (\bar{u}_2)_{\max}] \times [0, T]$. In the computational domain, a uniform difference grid is set in time $t$ and in spatial coordinates $\bar{u}_1$ and $\bar{u}_2$:

$$(\bar{u}_1)_j = j\Delta \bar{u}_1, \qquad j \in [1, M], \qquad (\bar{u}_1)_{\min} = (\bar{u})_{j=1}, \qquad (\bar{u}_1)_{\max} = (\bar{u})_{j=M},$$

$$(\bar{u}_2)_k = k\Delta \bar{u}_2, \qquad k \in [1, L], \qquad (\bar{u}_2)_{\min} = (\bar{u})_{k=1}, \qquad (\bar{u}_2)_{\max} = (\bar{u})_{k=L},$$

$$t_n = n\Delta t, \qquad n = 0, 1, 2, \dots T/\Delta t - 1,$$

where $\Delta \bar{u}_1$ and $\Delta \bar{u}_2$ are steps in spatial coordinates, and $\Delta t$ is a step in time.

2. In these notations, for the constructed grid, we have the following difference equation for the PDEs (18) and (19):

$$P_{j,k}^{n+1} = P_{j,k}^n + r_1\left[P_{j+1,k}^n - 2P_{j,k}^n + P_{j-1,k}^n\right] + r_2\left[P_{j,k+1}^n - 2P_{j,k}^n + P_{j,k-1}^n\right] + 4\Delta t P_{j,k}^n +$$

$$\frac{\Delta t}{2\Delta \bar{u}_1}\left[(\bar{u}_1^2)_j - (\bar{u}_2^2)_k + \Omega_{0(n)}^2\right]\left(P_{j+1,k}^n - P_{j-1,k}^n\right) + \frac{\Delta t}{\Delta \bar{u}_2}(\bar{u}_1)_j(\bar{u}_2)_k\left(P_{j,k+1}^n - P_{j,k-1}^n\right), \tag{87}$$

where the following notations are introduced:

$$P_{j,k}^n = P(j\Delta \bar{u}_1, k\Delta \bar{u}_2; n\Delta t), \quad \Omega_{0(n)}^2 = \Omega_0^2(t_n), \quad r_1 = \varepsilon^{(r)}\frac{\Delta t}{(\Delta \bar{u}_1)^2}, \quad r_2 = \varepsilon^{(i)}\frac{\Delta t}{(\Delta \bar{u}_2)^2}.$$

3. To calculate Equations (18) and (19), it is also necessary to set two boundary conditions in the form of difference equations on the coordinate axes $\bar{u}_1$ and $\bar{u}_2$, respectively, which can be easily found by approximating Equation (87). Note that these difference equations must be solved taking into account the Dirichlet boundary conditions; $\mathbb{P}_l(x)|_{x = \pm |s|} = 0$, $(|s| \gg 1)$, where the index $(l = 1, 2)$ denotes the first and second boundary conditions, respectively.

4. The condition is set at the center of the coordinate axes:

$$P(\bar{u}_1, \bar{u}_2 = 0, t)\big|_{\bar{u}_1 = 0} = P(\bar{u}_1 = 0, \bar{u}_2, t)\big|_{\bar{u}_2 = 0}.$$

In addition, the Dirichlet condition $P(\bar{u}_1, \bar{u}_2, t)\big|_{\partial \mathcal{G}} = 0$ is specified on the boundaries of the computational domain, where $\partial \mathcal{G}$ denotes the boundary.

5. As an initial condition, instead of the Dirac delta function, we use the Gaussian distribution:

$$P(\bar{u}_1, \bar{u}_2, t)\big|_{t=0} \approx \sigma \exp\left\{-\omega\left(\left[\frac{\bar{u}_1}{a}\right]^2 + \left[\frac{\bar{u}_2}{b}\right]^2\right)\right\}.$$

Note that the parameters $\sigma = 500$, $\omega = \pi\sigma a^2$ and $a = b = 1/2$ included in the function were chosen in such a way to normalize the initial distribution to unit.

In this case, the following inequalities must hold:

$$r_1 + r_2 \le 1/2, \qquad C_1 + C_2 \le 1.$$

Such necessary conditions are quite sufficient in determining the upper limit of the time step $\Delta t_{crit}$: $\Delta t \le \Delta t_{crit}$ for all options presented in the table below. To estimate the sufficient stability condition, we can use the approximation taken from the one-dimensional case:

$$C_i^2 \le r_i \le 1, \qquad i = 1, 2.$$

In real test calculations, the original differential equation behaves more complicated than the model linear equation. However, the real $\Delta t_{crit}$ obtained from the calculations is close to the value determined from the given difference scheme stability inequalities.

Formally, we can represent the system of Equations (87) and (88) in matrix form, which in the general case looks like:

$$\left(f_{j,k}^{n+1}\right)_l = A_l f_{j,k}^n, \qquad l = 1, 2, 3,$$

where $A_l$ denotes the transition matrix for the variable $f$, in which connection $A_1$ is the transition matrix for the variable $\left(f_{j,k}^{n+1}\right)_1 = P$ (see Equation (87)), and the matrices $A_2$ and $A_3$ are for the variables $\left(f_{j,k}^{n+1}\right)_2 = Q^{(i)}$ and $\left(f_{j,k}^{n+1}\right)_3 = Q^{(r)}$ (see Equation (88)). To save space, we do not write out these matrices in detail. It is important to note that the calculation process itself is clear from Equations (87) and (88).

**Listing 2.** Numerical algorithm for solving the system of PDEs (52) after coordinate transformation.

1.  The continuous region $\mathbb{R}^2$ for the PDEs system (66) is replaced by a discrete grid, as described in Listing 1.
2.  Using the PDEs system (66), we can obtain the following system of difference equations on the constructed grid:

$$[Q^{(i)}]_{j,k}^{n+1} = [Q^{(i)}]_{j,k}^n + r_1 \left\{ [Q^{(i)}]_{j+1,k}^n - 2[Q^{(i)}]_{j,k}^n + [Q^{(i)}]_{j-1,k}^{n+1} \right\} + r_2 \left\{ [Q^{(i)}]_{j,k+1}^n - \right.$$

$$\left. 2[Q^{(i)}]_{j,k}^n + [Q^{(i)}]_{j,k-1}^{n+1} \right\} + \frac{\Delta t}{2\Delta \bar{u}_1} \left[ (\bar{u}_1^2)_j - (\bar{u}_2^2)_k + \Omega_{0(n)}^2 \right] \left\{ [Q^{(i)}]_{j+1,k}^n - [Q^{(i)}]_{j-1,k}^n \right\}$$

$$+ \frac{\Delta t}{\Delta \bar{u}_2} (\bar{u}_1)_j (\bar{u}_2)_k \left\{ [Q^{(i)}]_{j,k+1}^n - [Q^{(i)}]_{j,k-1}^n \right\} + \Delta t \left\{ 5(\bar{u}_1)_j [Q^{(i)}]_{j,k}^n + (\bar{u}_2)_k [Q^{(r)}]_{j,k}^n \right\},$$

$$[Q^{(r)}]_{j,k}^{n+1} = [Q^{(r)}]_{j,k}^n + r_1 \left\{ [Q^{(r)}]_{j+1,k}^n - 2[Q^{(r)}]_{j,k}^n + [Q^{(r)}]_{j-1,k}^{n+1} \right\} + r_2 \left\{ [Q^{(r)}]_{j,k+1}^n - \right.$$

$$\left. 2[Q^{(r)}]_{j,k}^n + [Q^{(r)}]_{j,k-1}^{n+1} \right\} + \frac{\Delta t}{2\Delta \bar{u}_1} \left[ (\bar{u}_1^2)_j - (\bar{u}_2^2)_k + \Omega_{0(n)}^2 \right] \left\{ [Q^{(r)}]_{j+1,k}^n - [Q^{(r)}]_{j-1,k}^n \right\}$$

$$+ \frac{\Delta t}{\Delta \bar{u}_2} (\bar{u}_1)_j (\bar{u}_2)_k \left\{ [Q^{(r)}]_{j,k+1}^n - [Q^{(r)}]_{j,k-1}^n \right\} + \Delta t \left\{ 5(\bar{u}_1)_j [Q^{(r)}]_{j,k}^n - (\bar{u}_2)_k [Q^{(i)}]_{j,k}^n \right\},$$

$$(88)$$

where $\left[Q^{(v)}\right]_{j,k}^n = Q^{(v)}(j\Delta \bar{u}_1, k\Delta \bar{u}_2; n\Delta t)$; in addition, $v = (i, r)$.

3.  Similarly, as in Listing 1, for the solutions $Q^{(i)} \bar{u}_1, \bar{u}_2, t)$ and $Q^{(r)}(\bar{u}_1, \bar{u}_2, t)$, boundary conditions are set on the $\bar{u}_1$ and $\bar{u}_2$ axes in the form of difference equations, which can be obtained by approximating Equation (88) on the same axes. Note that we solve each equation obtained for the boundary conditions as an internal Dirichlet problem with a zero value of the solution at the boundary.
4.  As in the case of the probability density $P(\bar{u}_1, \bar{u}_2, t)$ (see Listing 1), the solutions $Q^{(i)}(\bar{u}_1, \bar{u}_2, t)$ and $Q^{(r)}(\bar{u}_1, \bar{u}_2, t)$ are subject to similar conditions at the center of the coordinate axes:

$$Q^{(i)}(\bar{u}_1, \bar{u}_2 = 0, t)\big|_{\bar{u}_1=0} = Q^{(i)}(\bar{u}_1 = 0, \bar{u}_2, t)\big|_{\bar{u}_2=0},$$

$$Q^{(r)}(\bar{u}_1, \bar{u}_2 = 0, t)\big|_{\bar{u}_1=0} = Q^{(r)}(\bar{u}_1 = 0, \bar{u}_2, t)\big|_{\bar{u}_2=0}.$$

In addition, we will assume that at the boundary of the computational domain, the solutions are subject to the following conditions:

$$Q^{(i)}(\bar{u}_1, \bar{u}_2, t)\big|_{\partial \mathcal{G}} = 0, \qquad Q^{(r)}(\bar{u}_1, \bar{u}_2, t)\big|_{\partial \mathcal{G}} = 0.$$

5.  Finally, as an initial condition for solving the system of Equations (88) for both solutions $Q^{(i)}(\bar{u}_1, \bar{u}_2, t)$ and $Q^{(r)}(\bar{u}_1, \bar{u}_2, t)$, Gaussian distribution with parameters as in Listing 1 is chosen.

*7.1. Distributions of the Free Environmental Fields*

To calculate the distribution equation for the free fields of the environment (18) and (19), we use the difference Equation (87), which is described in detail in Listing 1. For the numerical simulation of the problem, the initial data given in Table 1 were used.

**Table 1.** Initial data required for problem modeling.

| | | | |
|---|---|---|---|
| $\nu = 0.5;$ | $\gamma = 2;$ | $\varepsilon^{(r)} = 0.01;$ | $\varepsilon^{(i)} = 0.01;$ |
| $\nu = 0.5;$ | $\gamma = 2;$ | $\varepsilon^{(r)} = 1.00;$ | $\varepsilon^{(i)} = 0.01;$ |
| $\nu = 0.5;$ | $\gamma = 2;$ | $\varepsilon^{(r)} = 1.00;$ | $\varepsilon^{(i)} = 0.50.$ |

Note that each line of this table characterizes different states of the environment:

a. When the processes going on in the environment, both elastic and inelastic, are weak;

b. When elastic processes are strong and inelastic processes are weak;

c. When both elastic and inelastic processes are strong in the environment.

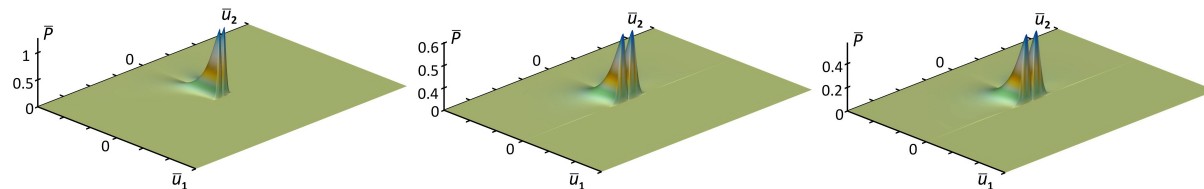

**Figure 1.** The different stages of the evolution of the free fields of the environment, respectively, at $t_1 = 1.5$, $t_2 = 10$ and $t_3 = 20$. Note that these distributions were calculated using the data from the first row of the table, which corresponds to weak elastic and inelastic processes in the environment. Comparing the distributions at different times, it is easy to see that as $t \sim 10$, the distribution $\bar{P}(u_1, u_2, t)$ normalized to unity is established or tends to its stationary limit.

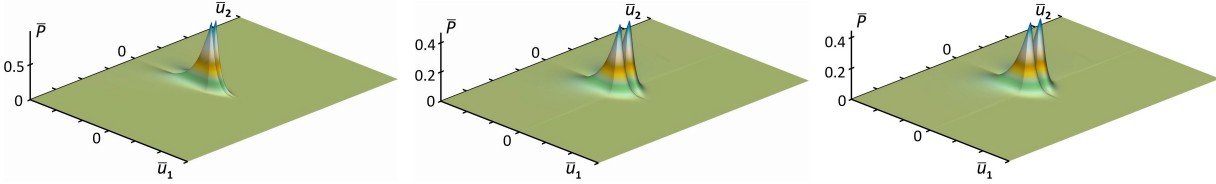

**Figure 2.** A series of graphs of the distribution of free fields of the environment at time points $t_1 = 1.5$, $t_2 = 10$, $t_3 = 20$. Recall that we used the data of the second line table, which characterizes strongly elastic and weakly inelastic processes occurring in the environment. An analysis of the graphs shows that the distribution tends to the stationary limit already at $t \sim 7$.

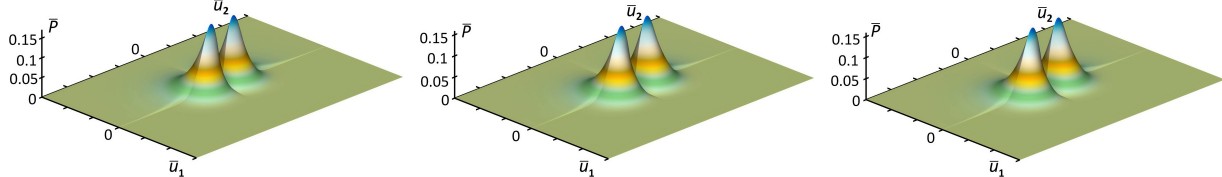

**Figure 3.** Distributions of free fields of the environment, respectively, at the time points $t_1 = 1.5$, $t_2 = 3$, $t_3 = 10$ and $t_4 = 20$. The data of the third row table, corresponding to strong elastic and inelastic processes occurring in the environment, were used for the calculation. As can be seen from the figures, the greater the constants that determine the powers of elastic and inelastic processes, the faster the distribution of environmental fields is established.

### 7.2. Distributions of Environmental Fields Taking into Account the Influence of the Oscillator

We now present figures illustrating the evolution of the normalized functions $\bar{Q}^{(r)}(u_1, u_2, t)$ and $\bar{Q}^{(i)}(u_1, u_2, t)$ depending on time. Using the algorithm developed in Listing 1, we can calculate and visualize all of these solutions. Below are the normalized distributions for three different cases.

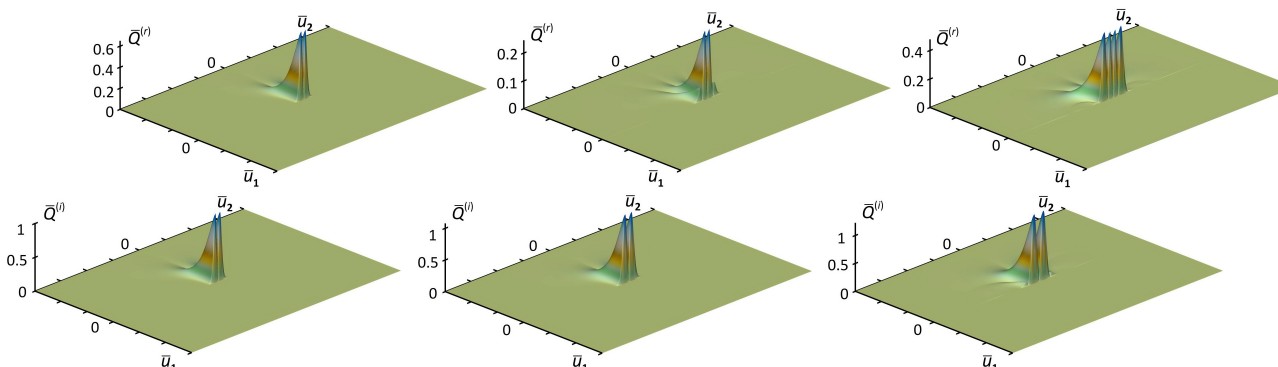

**Figure 4.** Distributions of the environment fields in the process of evolution, respectively, at the time points $t_1 = 1.5$, $t_2 = 10$ and $t_3 = 20$, calculated from the data of the first line table.

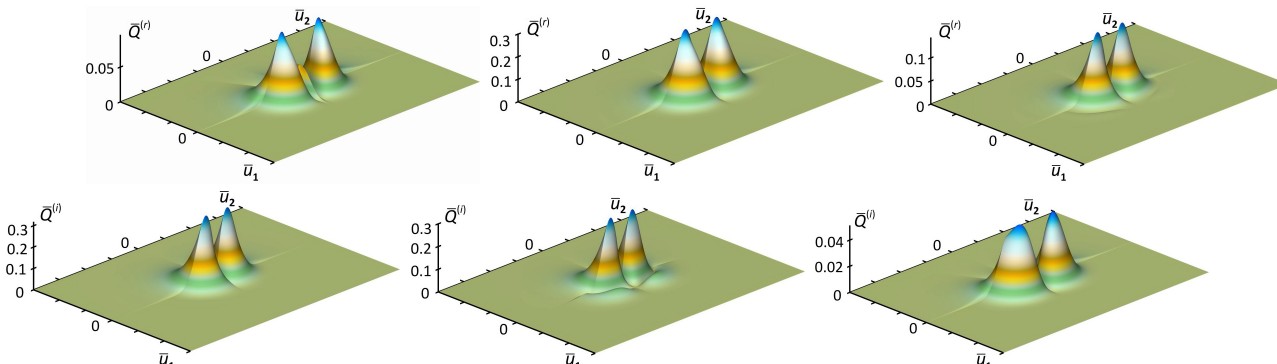

**Figure 5.** Distributions of the fields of the environment in the course of evolution, respectively, at the time points $t_1 = 1.5$, $t_2 = 10$ and $t_3 = 20$, calculated using the third line of the table.

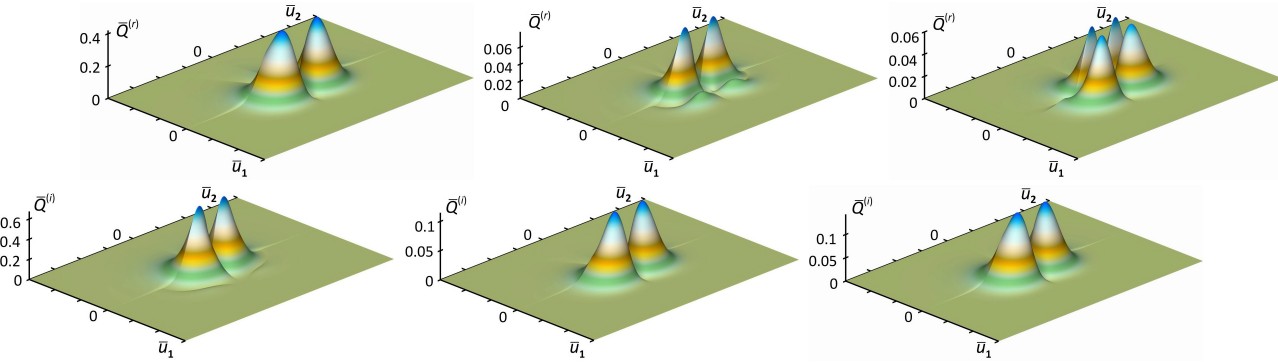

**Figure 6.** Distributions of environmental fields in the process of evolution, respectively, at time points $t_1 = 1.5$, $t_2 = 10$ and $t_3 = 20$. The calculations are performed for the case $\varepsilon^{(r)} = \varepsilon^{(i)} = 1$, when processes, both elastic and inelastic, are strongly developed.

In particular, as we see from these figures, with an increase in the constants of interactions with the environment, the time for establishing distributions is reduced.

### 7.3. Mathematical Expectation of the Oscillator Trajectory

Using all the above calculations, one can simulate the expected value of the oscillator trajectory in the absence of an external field based on Equation (35). We modeled the expected oscillator trajectory for both its real and imaginary parts for five different environmental states and visualized their behavior as a function of time (see Figure 7). In particular, as can be seen from the figures, in all the cases under consideration, the mathematical expectation of the trajectory not only has a non-trivial oscillatory character, but it decreases with time, taking both positive and negative values.

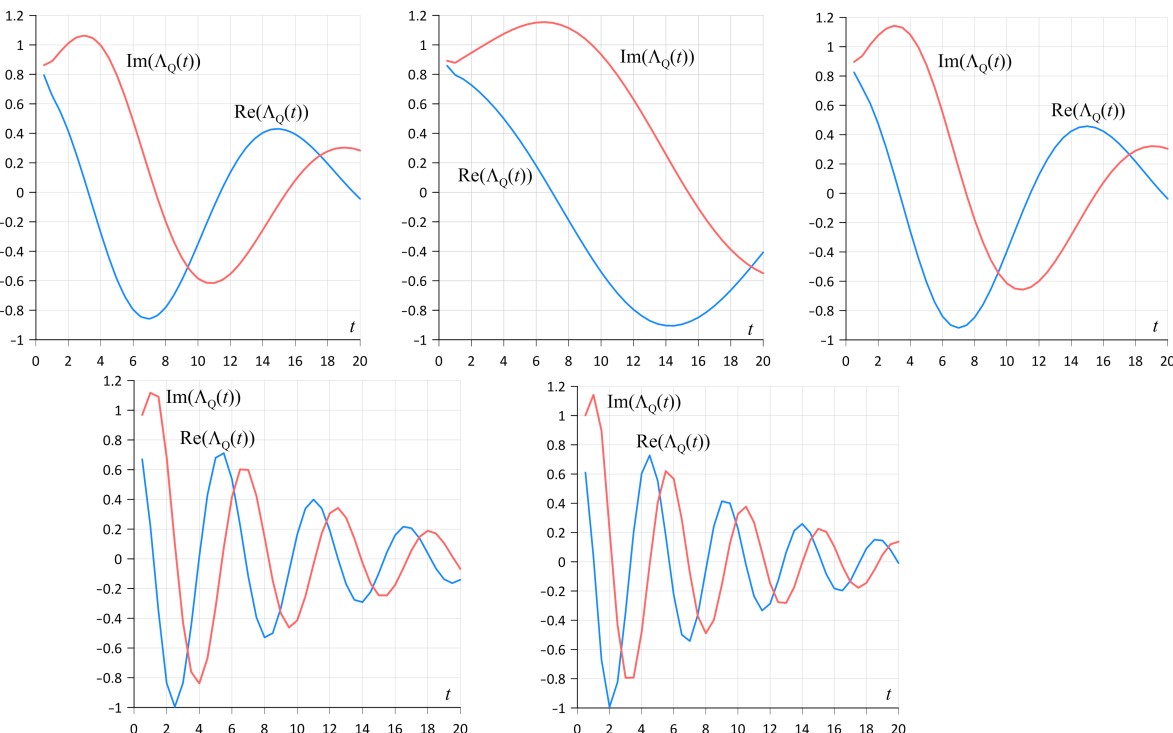

**Figure 7.** Mathematical expectations of the real (blue) and imaginary (red) parts of the oscillator trajectory for five sets of parameters. Left to right, front row: $(\varepsilon^{(r)} = \varepsilon^{(i)} = 0.01)$, $(\varepsilon^{(r)} = 1, \varepsilon^{(i)} = 0.001)$ and $(\varepsilon^{(r)} = 1, \varepsilon^{(i)} = 0.01)$; in the second row: $(\varepsilon^{(r)} = 1, \varepsilon^{(i)} = 0.5)$ and $(\varepsilon^{(r)} = \varepsilon^{(i)} = 1)$.

### 7.4. Calculation of Topological and Geometric Features of the Manifold $\Sigma_{\mathbf{u}}^{(2)}(t)$

As we saw above, the off-diagonal term of the metric tensor $y = g^{12}(u_1, u_2, t)$ of a manifold $\Sigma_{\mathbf{u}}^{(2)}(t)$ is determined by the algebraic equation of the 4th degree (64) with coordinates- and time-dependent coefficients. Computing this equation with the Mathematics–Wolfram solver for three sets of parameters (see data of the table), we obtain sets of surfaces for the off-diagonal term of the metric tensor (see Figures 8–11), which allows us to study and understand the geometric and topological features of the manifold $\Sigma_{\mathbf{u}}^{(2)}(t)$. An analysis of the surfaces (see Figure 8) shows that when the oscillator is immersed in an environment with weak elastic and inelastic interactions, these surfaces do not have interesting topological features. However, there is an obvious singularity at the point $u_1 = 0 - \epsilon$, $(u_1 \in (-\infty, 0] \setminus 0, \epsilon > 0)$, because as $\epsilon \to 0$, the term of $y = g^{12}(u_1, u_2, t)$ tends to plus infinity in the upper half-space when as in the lower half-space, the term $y = -g^{21}(u_1, u_2, t)$ tends to minus infinity. It should be noted that in the case under consideration, the evolution of the environment characteristically does not change the geometry of space but only shifts the minimum point of the surface.

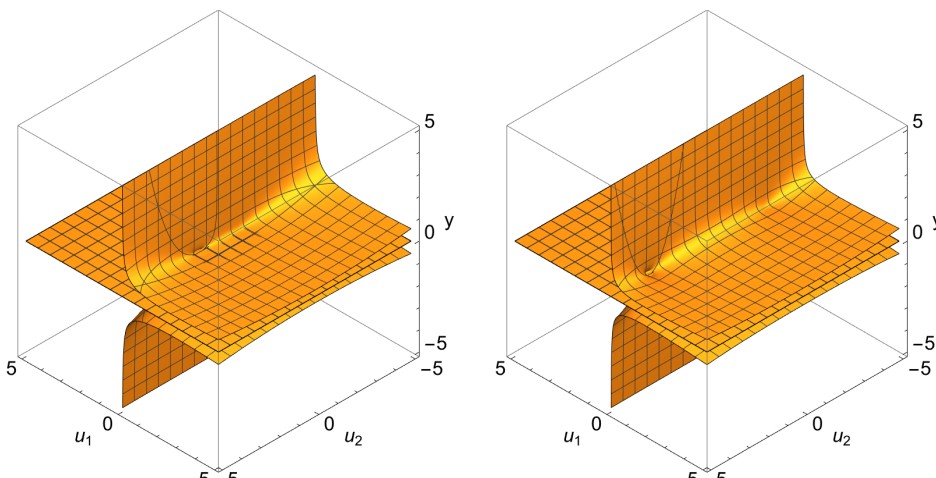

**Figure 8.** The two figures (from left to right) show the evolution of the space $\Sigma_{\mathbf{u}}^{(2)}(t)$ from the asymptotic state ($in$) with the environment data $\left(\varepsilon^{(r)} = \varepsilon^{(i)} = 0.01\right)$ and frequency $\Omega_0^- = 1$, to the asymptotic state ($out$) with frequency $\Omega_0^+ = 3$.

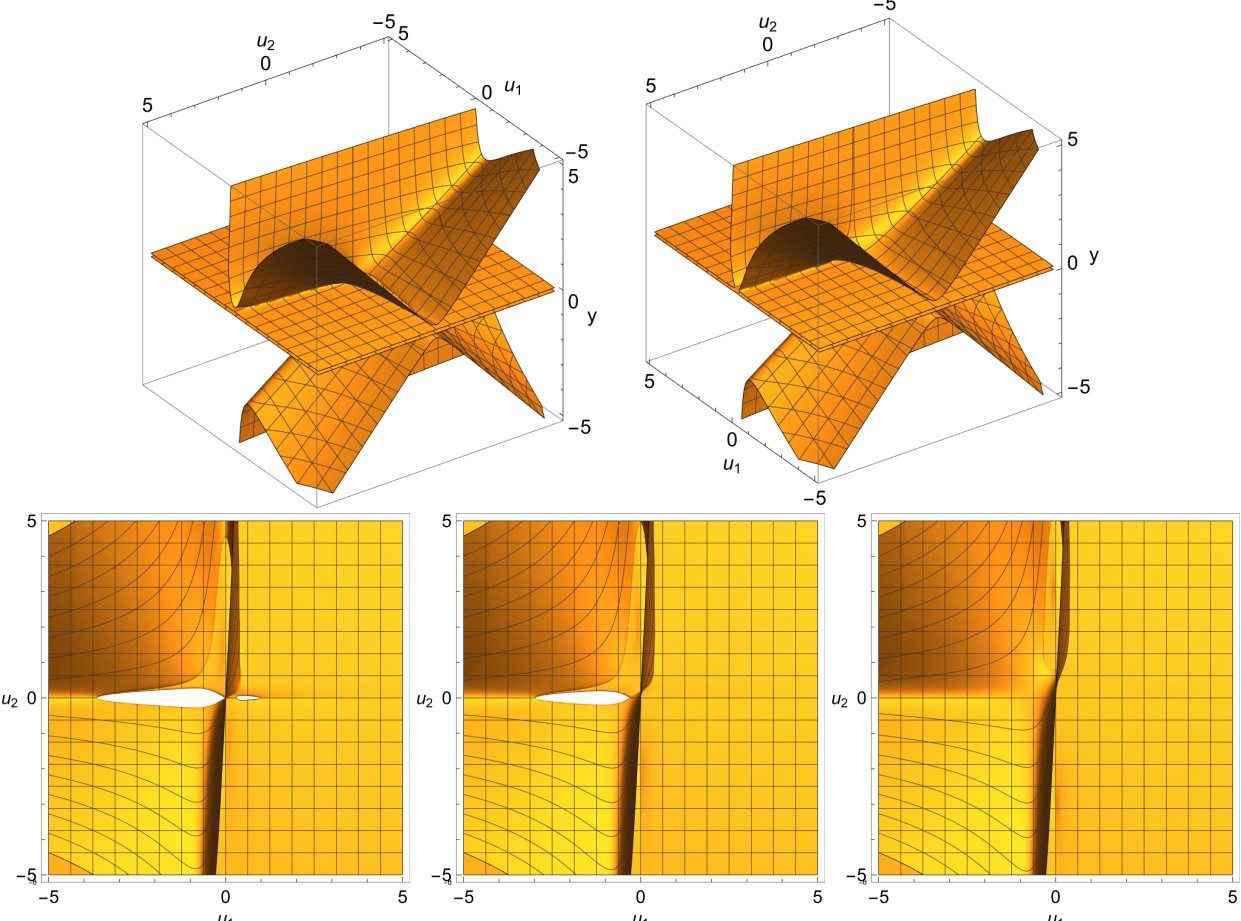

**Figure 9.** The first row on the right shows a topological space with the first Betti number equal to 2 in the ($in$) asymptotic state, which in the process of evolution to the ($out$) asymptotic state transits into two simply connected half-spaces with displaced holes. The second row shows three two-dimensional pictures—projections of the manifold $\Sigma_{\mathbf{u}}^{(2)}(t)$ onto the plane $(u_1, u_2)$, characterizing the evolution of the topological singularities of the manifold when transitioning between these asymptotic states defined by the data $(\Omega_0^- = \Omega_0(t) = 1 \ \Omega_0(t) = 2, \ \Omega_0^+ = \Omega_0(t) = 3)$ and $(\varepsilon^{(r)} = 1, \ \varepsilon^{(i)} = 0.01)$, respectively.

In the presence of strong elastic and weak inelastic processes in the environment, the calculations lead to the formation of the following surfaces (see Figure 9). An analysis of the pictures shows that the arising manifold is topological space with the first Betti number is equal to 2, which during evolution transits to the manifold consisting from two sub-manifolds each of which is a simply connected topological space with spatially shifted typological singularities. Moreover, the shift between the topologies of the upper and lower sub-manifolds is present already in the $(in)$ asymptotic state. In the course of evolution, this shift only increases, while the gap intersection decreases and already in the $(out)$ state becomes equal to zero. Moreover, the shift between the topologies of the upper $\Sigma_{\mathbf{u}}^{(2)+}(t)$ and lower $\Sigma_{\mathbf{u}}^{(2)-}(t)$ sub-manifolds is present in the $(in)$ asymptotic state (this can be verified by analyzing three-dimensional graphs in Figure 9). In the course of evolution, this shift only increases, while the gap intersection decreases and already in the $(out)$ state becomes equal to zero. In other words, the manifold $\Sigma_{\mathbf{u}}^{(2)}(t)$ can be represented as the union of two sub-manifolds:

$$\Sigma_{\mathbf{u}}^{(2)}(t) \cong \Sigma_{\mathbf{u}}^{(2)+}(t) \sqcup \Sigma_{\mathbf{u}}^{(2)-}(t), \tag{89}$$

where $\Sigma_{\mathbf{u}}^{(2)+}(t)$ denotes a submanifold of the upper half-space, while $\Sigma_{\mathbf{u}}^{(2)-}(t)$ is a submanifold of the lower half-space. As we will see below, this manifold can evolve into a disjoint union of two sub-manifolds in the course of evolution.

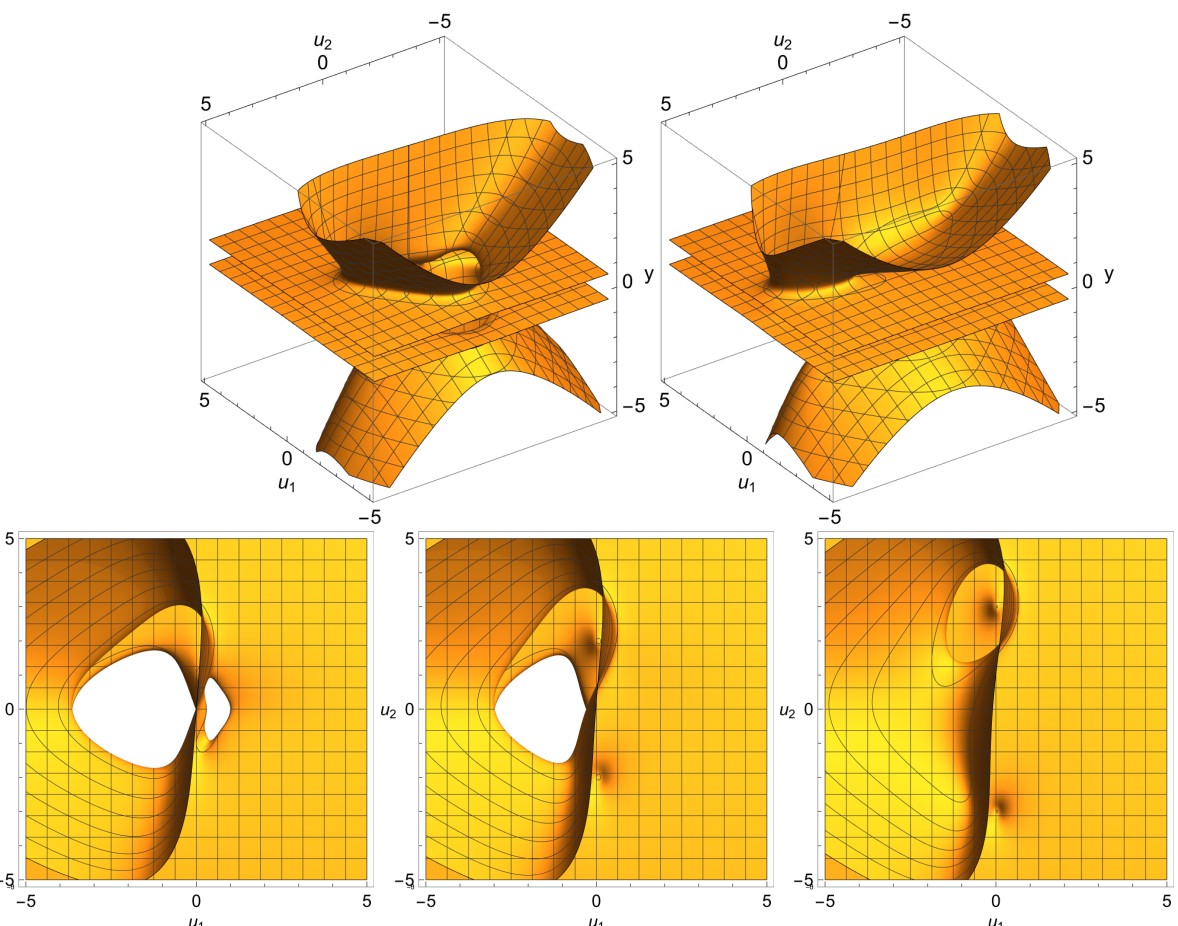

**Figure 10.** The two figures in the first row (from left to right) show the evolution of the manifold $\Sigma_{\mathbf{u}}^{(2)}(t)$ from the asymptotic state $(in)$ with data $\left(\Omega_0^- = 1, \varepsilon^{(r)} = 1, \varepsilon^{(i)} = 0.5\right)$ to the $(out)$ asymptotic state with data $\left(\Omega_0^+ = 3, \varepsilon^{(r)} = 1, \varepsilon^{(i)} = 0.5\right)$. The second row represents two-dimensional projections of this manifold onto the $(u_1, u_2)$ plane, showing the evolution of its topological features.

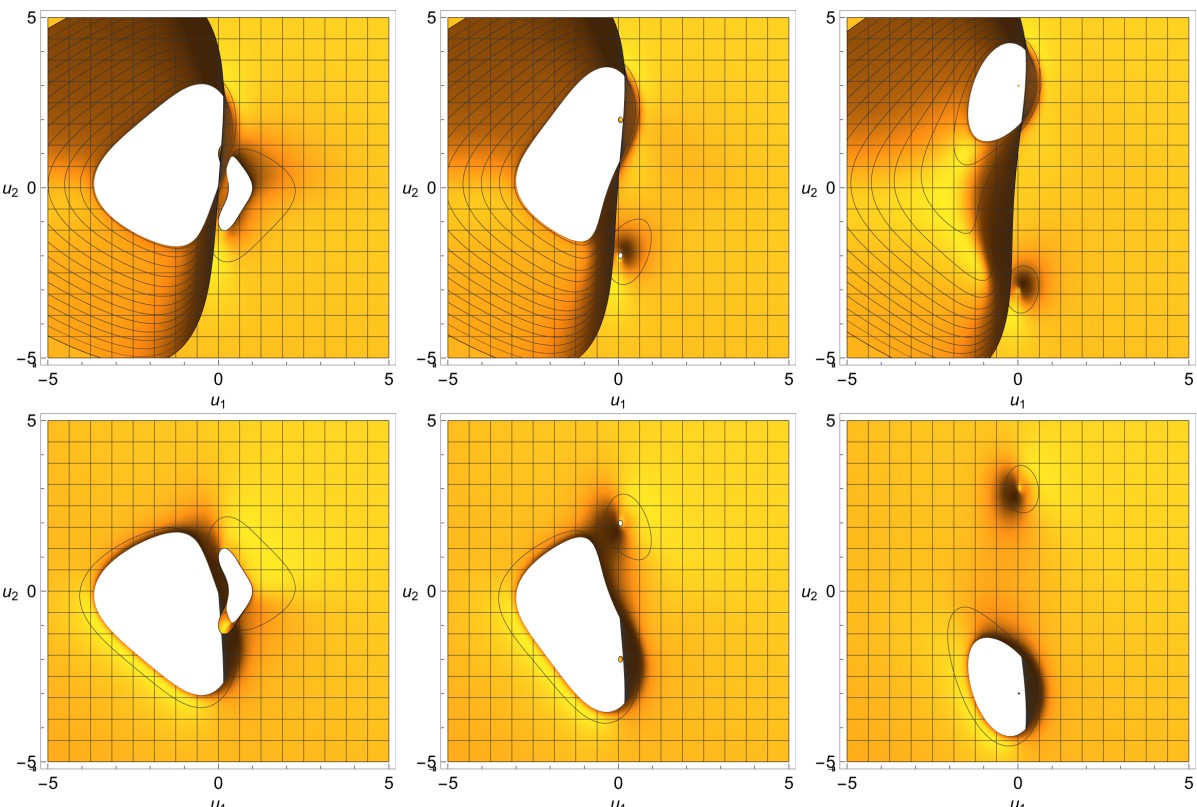

**Figure 11.** The three figures in the first row (from left to right) show the evolution of the topological singularities of the submanifold $\Sigma_{\mathbf{u}}^{(2)+}(t)$ from the asymptotic state ($in$) with data $\left(\Omega_0^- = 1, \varepsilon^{(r)} = 1, \varepsilon^{(i)} = 0.5\right)$, to ($out$) an asymptotic state where the frequency is $\Omega_0^+ = 3$. In the second row (from left to right), there are figures showing a similar evolution for the sub-manifold $\Sigma_{\mathbf{u}}^{(2)-}(t)$.

Finally, let us consider the case when strong elastic and non-weak inelastic processes occur in the environment. As calculations show (see Figure 10), in this case, the manifold also has the first Betti nuber equal to 2 but with different sizes and slot configurations. In particular, the first row in Figure 10 shows three-dimensional visualizations of the elements of the metric tensor $g^{12}(u_1, u_2, t)$ and $g^{21}(u_1, u_2, t)$ as functions of coordinates $(u_1, u_2)$ in asymptotic states ($in$) and ($out$), respectively. The second row (Figure 10) shows the graphs showing the evolution of the topological features of the manifold $\Sigma_{\mathbf{u}}^{(2)}(t)$ in the transition from ($in$) to the state ($out$). As can be seen from the figures, the manifold $\Sigma_{\mathbf{u}}^{(2)}(t)$ loses its topological features in the process of evolution. However, since we know that the manifold $\Sigma_{\mathbf{u}}^{(2)}(t)$ is the union of two sub-manifolds (see (89)), a natural question arises: do these sub-manifolds retain any of their topological singularities during evolution? As the calculations and their two-dimensional visualizations (Figure 11) show, each of the sub-manifolds (see the first and second rows of the graphs) in the process of evolution passes from a topological space with the first Betti number equal to 2 to a simply connected topology. However, as we can see, these topologies are displaced and do not have a common hole (see the graphs in the second line of Figure 10, obtained by combining the graphs of the first two lines of Figure 11). It is important to note that depending on the parameters, a topological manifold with the first Betti number equal to 3 also arises, which is a union of two oriented topological submanifolds $\Sigma_{\mathbf{u}}^{(2)+}(t)$ and $\Sigma_{\mathbf{u}}^{(2)-}(t)$ with the first Betti number equal to 3. In the end, we note that the manifold $\Sigma_{\mathbf{u}}^{(2)}(t)$, regardless of the state of the environment, at the end of its evolution in the state ($out$) is a disjoint union of two sub-manifolds.

### 7.5. Features of Entropy Calculation

The integrals in the equations for the ordinary (84) and generalized (86) entropies were calculated using the trapezoidal method.

However, due to the fact that in the expressions for the entropy, there is a term of type $A\ln(A)$, which has a singularity at $A = 0$, the condition $A\ln(A) = 0$ is introduced, which makes it possible to eliminate this singularity. We performed entropy calculations using Formulas (84) and (86) for three different states of the environment and visualized them (see Figure 12) taking into account the data of Table 1. As follows from these graphs, the usual entropy $\mathcal{S}(t)$ (left column) in the first two cases continuously increases with time and reaches a constant value in the $(out)$ state. In the third case, when inelastic precessions in the environment are strong enough, as we observe, from some moment, the increase in entropy passes into the stage of its decrease, and already at the long times, when the system goes into $(out)$ state, it takes a constant value. The right column contains graphs of the generalized entropy $\mathcal{S}_{gen}(t)$, which, in addition to being non-monotonic, also contain areas with negative values. This behavior of entropy is quite explainable for a closed self-organizing system, which at some point can generate negentropy to stabilize the state of the joint system. In particular, the negative value of entropy can also be explained by the short-term capture or "swallowing" of a small or oscillator subsystem by a large subsystem or thermostat.

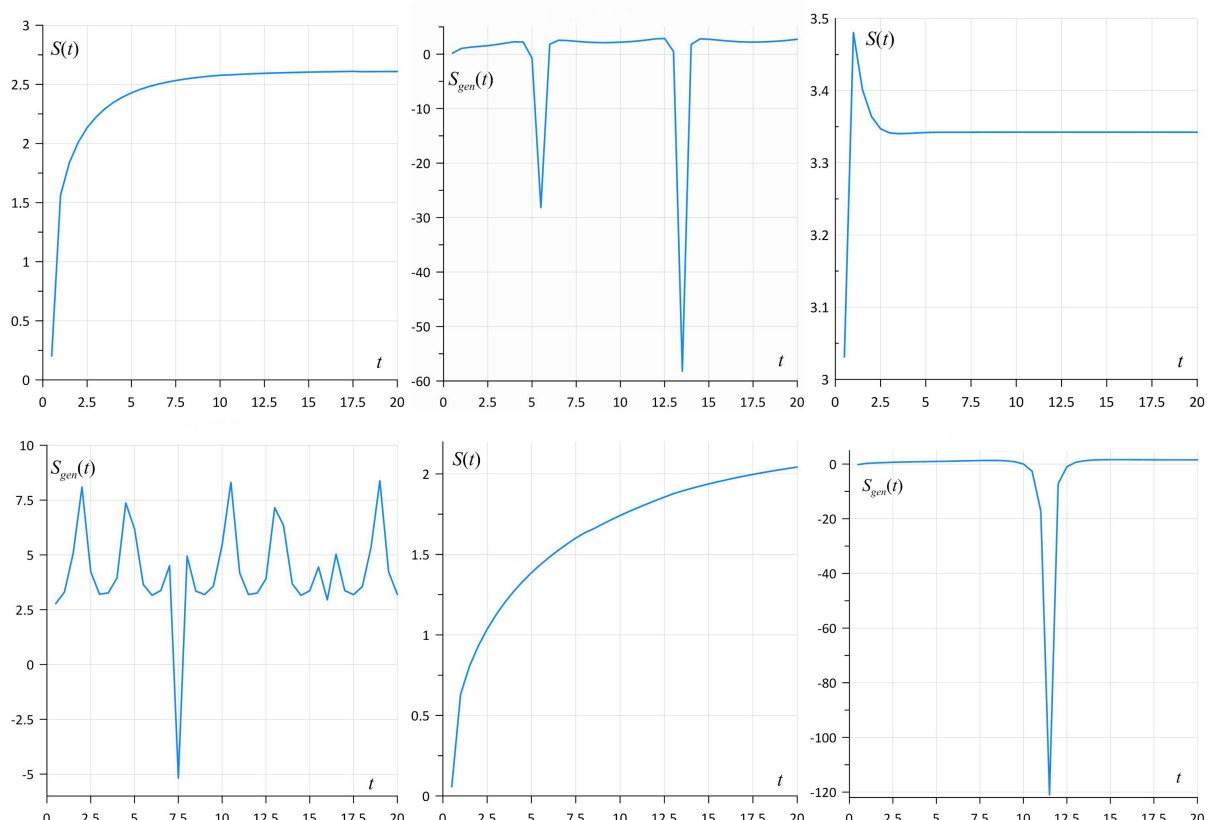

**Figure 12.** In the figure, the left column of three graphs describes the Shannon entropy for three different environmental states, while the right column describes the generalized entropy describing the JS for the same environmental states.

As can be seen from the graphs of the right column, in the first two cases, the generalized entropy $\mathcal{S}_{gen}(t)$ relatively fast tends a constant value, while in the third case, the process of stabilization of the combined system proceeds non-monotonically and takes a long time.

## 8. Conclusions

The main achievement of this work is the development of a mathematically rigorous representation that allows one to study the statistical properties of a classical oscillator with its random environment as a problem of self-organization of a closed self-consistent system. Note that such a statement in the philosophical sense corresponds to the consideration of the problem within the framework of Plato's concept, which excludes the loss of information in the JS. We have carried out a mathematical implementation of this idea within the framework of a complex probabilistic process that satisfies the SDE of the Langevin type. The paper considers three typical scenarios of a random environment or thermostat, for which, in the limit of statistical equilibrium, the kinetic equations for the distribution of the fields of the environment are derived (see Equations (18), (19), (25) and (26)). With the help of these equations, the measures of functional spaces are determined, which make it possible in the future to build mathematical expectations of the corresponding physical parameters of the problem. We use the generalized Feynman–Kac theorem [18,19] to compactify the infinite-dimensional functional integral describing the expectation of the oscillator trajectory and reduce it to the two-dimensional integral representation, where the integrand is the solution of the complex second-order PDE given on a two-dimensional manifold $\Sigma_{\mathbf{u}}^{(2)}(t)$.

The second important result of the work is the proof that the subspace $\Sigma_{\mathbf{u}}^{(2)}(t)$ is generally described by a non-commutative geometry, which may also have important geometric and topological features. In the case of non-intensive random processes in the environment, i.e., when $(\varepsilon^{(r)}, \varepsilon^{(i)}) \ll 1$, the manifold $\Sigma_{\mathbf{u}}^{(2)}(t) \cong \mathbb{E}_{\mathbf{u}}^{(2)+}(t) \sqcup \mathbb{E}_{\mathbf{u}}^{(2)-}(t)$, where $\mathbb{E}^{(2)}{}_{+\mathbf{u}}(t)$ and $\mathbb{E}_{\mathbf{u}}^{(2)-}(t)$ are Euclidean subspaces with one singular boundary (see Figure 8). As the power of random processes in the environment $(\varepsilon^{(r)}, \varepsilon^{(i)}) \sim 1$ increases, both the geometric properties of the manifold $\Sigma_{\mathbf{u}}^{(2)}(t) \cong \Sigma_{\mathbf{u}}^{(2)+}(t) \sqcup \Sigma_{\mathbf{u}}^{(2)-}(t)$ and their topological features change strongly. In particular, as shown in Figure 10, the sub-manifolds $\Sigma_{\mathbf{u}}^{(2)+}(t)$ and $\Sigma_{\mathbf{u}}^{(2)-}(t)$ have a non-Euclidean curvilinear geometry and topologies with the first Betti number (see for example [13]) equal to 2, which, in the $(out)$ state, go over to a simply connected topology. In other words, conformational transformations of an additional subspace $\Sigma_{\mathbf{u}}^{(2)}(t)$ of a self-organizing classical system, taking into account its geometric and typological features, lead to radical differences in the description of the dynamics of a classical system without an environment and when it is immersed in a random environment. However, it is obvious that the correct formulation is to solve the problem not on the two-dimensional Euclidean space $\mathbb{R}^2$ but on the manifold $\Sigma_{\mathbf{u}}^{(2)}(t)$, which is the union of two topological sub-manifolds $\Sigma_{\mathbf{u}}^{(2)+}(t)$ and $\Sigma_{\mathbf{u}}^{(2)-}(t)$, respectively. We emphasize once again that this is due to the fact that the generated manifold $\Sigma_{\mathbf{u}}^{(2)}(t)$ is generally described by a non-commutative geometry that has non-trivial topological singularities. In this paper, we develop an efficient mathematical algorithm for the sequential and parallel calculation of various characteristic parameters of the problem, taking into account that the manifold $\Sigma_{\mathbf{u}}^{(2)}(t)$ is the Euclidean space. Note that this approximation takes place when the interaction constants of the oscillator with the environment are small.

In the near future, we plan to generalize the computational algorithm for performing calculations on just such a manifold. Recall that in this case, the distribution of the environmental fields will be described by the tensor Equations (49) and (50) where the off-diagonal element $g^{12}(u_1, u_2, t) = -g^{21}(u_1, u_2, t)$ will be determined by the algebraic equation of the 4th degree (64). The latter will allow numerical methods to study important features of the dynamics of a classical system within the framework of an ideologically more consistent and accurate Platonic concept.

In conclusion, it should be noted that the study of quantum analogues of the considered classical models, taking into account the non-commutativity of the emerging geometries, will be extremely interesting and rich in new and unexpected results.

**Author Contributions:** Conceptualization, A.S.G.; Formal analysis, V.V.M. and K.A.M.; Funding acquisition, A.V.B.; Investigation, A.S.G. and A.V.B.; Software, V.V.M.; Visualization, K.A.M. All authors have read and agreed to the published version of the manuscript.

**Funding:** Gevorkyan A.S. is grateful to grant N 21T-1B059 of the Science Committee of Armenia, which partially funded this work. The research was carried out with partial financial support from the Ministry of Science and Higher Education of the Russian Federation within the framework of the program "World-Class Research Center: Advanced Digital Technologies" (contract No. 075-15-2020-903 dated 16 November 2020).

**Data Availability Statement:** The data presented in this study are the model data. Data sharing is not applicable to this article.

**Conflicts of Interest:** The authors declare no conflict of interest.

## Abbreviations

The following abbreviations are used in this manuscript:

| | |
|---|---|
| SEq | Statistical Equilibrium |
| PDE | Partial Differential Equation |
| SE | Small Environment |
| JS | Joint System |
| SDE | Stochastic Differential Equation |

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
