# Peer review of "Theoretical and Numerical Study of Self-Organizing Processes in a Closed System Classical Oscillator and Random Environment"

_mathematics, doi:10.3390/math10203868_

Round 1
Reviewer 1 Report
Attached pdf file.

Author Response
Thank you very much for reading our large work carefully.

Reviewer 2 Report
This is a long paper (45 pages), but it is well-written and interesting. The authors study a self-organization system consisting of a classical oscillator in a random environment. This leads to a Langevin stochastic differential equation, and in the limit of statistical equilibrium to a Fokker-Planck equation (Theorem 1), which is then further studied in different scenarios. There are also connections to noncommutative geometry. The paper is well summarized in the introduction and in the final section "Conclusion". The introduction also gives historical backgrounds in a wider perspective. Here I think that there is at least one mistake: it is indicted that Democritus reached the age of one hundred years (460-360 BC), but according to my sources his life span should be 460-370 BC (good enough, but anyway).
Otherwise I am not able to check the paper in full detail, but generally speaking it looks good.
Author Response
Thank you very much for reading our paper carefully.

Reviewer 3 Report
In the limit of statistical equilibrium (SEq), second-order PDEs were derived. The mathematical expectation of
the oscillator trajectory was constructed in the form of a functional-integral representation. An algorithm for parallel modeling of the problem was developed. This work is interesting. However, some minor issues should be handled.
1. Are there any constraints between space-time grids?
2. FIG 1-3 should be Figs. 1-3. Authors should check these in the text.
3. On page 31, a title should be added in Table.
4. Authors should illustrate how to solve (85). A matrix form should preferably be provided.
Author Response
Thank you very much for reading our work carefully.

Round 2
Author Response
Dear reviewer, we made all necessary corrections in the text. You can see this in the pdf version of the manuscript, they are marked with bold. The text, as far as possible, was written in active voice.
